# Chromatin attachment to the nuclear matrix represses hypocotyl elongation in *Arabidopsis thaliana*

Linhao Xu [1,5], Shiwei Zheng[1,5], Katja Witzel[2], Eveline Van De Slijke[3,4], Alexandra Baekelandt[3,4], Evelien Mylle [3,4], Daniel Van Damme [3,4], Jinping Cheng[1], Geert De Jaeger[3,4], Dirk Inzé[3,4] & Hua Jiang [1] ✉

The nuclear matrix is a nuclear compartment that has diverse functions in chromatin regulation and transcription. However, how this structure influences epigenetic modifications and gene expression in plants is largely unknown. In this study, we show that a nuclear matrix binding protein, AHL22, together with the two transcriptional repressors FRS7 and FRS12, regulates hypocotyl elongation by suppressing the expression of a group of genes known as *SMALL AUXIN UP RNAs* (*SAURs*) in *Arabidopsis thaliana*. The transcriptional repression of *SAURs* depends on their attachment to the nuclear matrix. The AHL22 complex not only brings these SAURs, which contain matrix attachment regions (MARs), to the nuclear matrix, but it also recruits the histone deacetylase HDA15 to the *SAUR* loci. This leads to the removal of H3 acetylation at the *SAUR* loci and the suppression of hypocotyl elongation. Taken together, our results indicate that MAR-binding proteins act as a hub for chromatin and epigenetic regulators. Moreover, we present a mechanism by which nuclear matrix attachment to chromatin regulates histone modifications, transcription, and hypocotyl elongation.

The nuclear matrix, initially identified through the biochemical isolation of a nuclear fraction in eukaryotic cells in the 1970s[1], has been considered a crucial supporting scaffold for the organized compaction of genomic DNA. Despite its first discovery in mammals, the existence of the nuclear matrix in plants has been a subject of ongoing debate for decades[2]. The proposed structure of the nuclear matrix involves matrix attachment regions (MARs) on the chromatin and nuclear matrix binding proteins that attach to these MARs[3]. MARs are genomic elements that have been suggested to anchor chromatin to the nuclear matrix, contributing to the spatial organization of the genome. However, the functional relevance of the recruitment to the nuclear matrix, especially in plants, requires further exploration.

MARs are short DNA sequences in diverse eukaryotic nuclear genomes, including mammals and plants[4,5]. These DNA sequences are notable for their richness in AT sequences that likely narrow the minor DNA groove[6,7]. In mammals, MARs are essential for defining structural units of chromatin by binding to the nuclear matrix and organizing the chromatin into distinct loop domains[8–10], leading to positive or negative transcriptional effects[5]. In Arabidopsis (*Arabidopsis thaliana*), a preference for transcription start sites (TSS), enrichment of poly (dA:dT) tracts, and increased expression of nearby genes are observed for MARs identified on chromosome 4 using a tilling array[11]. Furthermore, genes containing MARs are prone to pronounced spatiotemporal expression regulation in Arabidopsis[12], suggesting the significant role of MAR attachment in transcriptional regulation. However, the distribution of MARs and their association with transcriptional outcomes in the entire genome, as well as the mechanisms by which nuclear matrix-associated

[1]Leibniz Institute of Plant Genetics and Crop Plant Research, Gatersleben 06466, Germany. [2]Leibniz Institute of Vegetable and Ornamental Crops, Großbeeren 14979, Germany. [3]Department of Plant Biotechnology and Bioinformatics, Ghent University, Ghent 9052, Belgium. [4]VIB Center for Plant Systems Biology, Ghent 9052, Belgium. [5]These authors contributed equally: Linhao Xu, Shiwei Zheng. ✉e-mail: jiangh@ipk-gatersleben.de

proteins regulate transcription, remain unclear in Arabidopsis and other plants.

MAR-binding proteins recruit MARs to the nuclear matrix[3]. One group of MAR-binding proteins contains an AT-hook motif, which can specifically bind to the AT-rich DNA sequence typical of MARs[13,14]. They also interact with other proteins and RNA to form larger complexes, serving as functional domains involved in multiple cellular processes such as transcriptional regulation, chromosome packaging, and development[3,15]. In mammals, the AT-hook motif-containing protein SATB1 (SPECIAL AT-RICH SEQUENCE-BINDING PROTEIN 1) is one of the most studied MAR-binding proteins essential for transcriptional regulation, chromatin organization, and histone modifications[16–21]. In Arabidopsis, a group of AHL (AT-HOOK MOTIF CONTAINING NUCLEAR-LOCALIZED) proteins has been shown to bind to nuclear matrix regions[22–24]. These AHLs are involved in many aspects of plant development, including hypocotyl elongation[25]. Higher-order mutants for various *AHL*s and the dominant-negative mutant allele of *AHL29*, *sob3-6* (*suppressor of phyB4 #3-6*), show a longer hypocotyl than wild type plants[25,26], potentially by suppressing the expression of auxin-related genes, such as *YUCCA8* (*YUC8*) and the *SMALL AUXIN UP-REGULATED RNA19* (*SAUR19*) subfamily[27]. The expression of *SAUR19* subfamily genes can also be affected by signaling from the steroid phytohormone brassinosteroid (BR), indicative for crosstalk between different signaling pathways involved in AHL-dependent hypocotyl regulation[28]. Additionally, overexpression of another AHL protein, AHLL22, inhibits hypocotyl elongation and delays the flowering time by influencing histone modification in Arabidopsis[29], suggesting that AHL22 is a potential interesting candidate to explore the function of MAR attachment. While it is known that phytohormone signaling pathways act downstream of AHLs in hypocotyl growth, a direct connection between MAR attachment to the nuclear matrix and gene expression, and how MAR attachment might influence gene expression are still largely unknown.

In this study, we used AHL22 as the target to explore how the AHL protein regulate gene expression and nuclear matrix attachment. We showed that AHL22 interacts with the transcriptional repressors FRS7 (FAR1-RELATED SEQUENCE 7) and FRS12 in the Arabidopsis nuclear matrix. AHL22, FRS7, and FRS12 cooperatively suppressed hypocotyl elongation by regulating, among others, *SAUR* expression levels. In line with the proposed function of MAR-binding proteins, AHL22, FRS7, and FRS12 were involved in MAR attachment, including the MARs present in or nearby multiple *SAUR* loci. While Arabidopsis MARs preferentially mapped to highly expressed genes, we also identified several examples in lowly expressed genes, indicating that MAR-binding proteins function in both activation and repression of gene expression. Furthermore,

the AHL22 complex recruited the histone deacetylase HDA15 at *SAUR* loci, leading to the silencing of *SAUR* genes and suppression of hypocotyl elongation. Taken together, our results reveal a novel mechanism by which nuclear matrix attachment of chromatin regulates histone modifications, transcription, and hypocotyl elongation.

## Results

### AHL22 interacts with FRS7 and FRS12 at the nuclear matrix

To understand how AHL22 regulates chromatin status, we identified interacting proteins using a tandem affinity purification (TAP) tagging strategy in an Arabidopsis PSB-D cell culture[30]. Among the proteins reproducibly identified in both replicates as interactors for AHL22, we noticed multiple AHLs, including AHL27 (also named ESCAROLA [ESC]) and AHL29 (also named SOB3) (Table 1, *SI Appendix*, Supplementary Data 1)[25]. In addition, we also identified the transcriptional repressor FRS12[31] (Table 1, *SI Appendix*, Supplementary Data 1). FRS12, together with its paralog FRS7, regulates flowering time and hypocotyl growth by repressing the expression of their target genes[31], similar to the known function of other AHL proteins in transcriptional silencing[27,28]. Therefore, we selected FRS7 and FRS12 for further analysis. To confirm the interaction between AHL22 and FRS7/12, we performed bimolecular fluorescence complementation (BiFC) assays. We detected an interaction between AHL22 and FRS7 or FRS12, as evidenced by the strong fluorescence signals from reconstituted nuclear yellow fluorescent protein (YFP) during the BiFc assay in tobacco (Fig. 1a). Förster resonance energy transfer measured by Fluorescence Lifetime Imaging Microscopy (FRET-FLIM) experiments further confirmed the interactions of AHL22-FRS7 and AHL22-FRS12 *in planta* (Fig. 1b, c). Finally, in a co-immunoprecipitation (Co-IP) assay, AHL22 was found to co-purify with FRS7 and FRS12, further demonstrating the in vivo interaction between AHL22 and FRS7/12 (Fig. 1d, e). Taken together, the above findings support the conclusion that AHL22 interacts with FRS7 and FRS12.

As AHLs have been proposed to be MAR-binding proteins[22–24,32], we hypothesized that AHL22 might interact with FRS7 and FRS12 at the nuclear matrix. To test this hypothesis, we isolated the Arabidopsis nuclear matrix fraction, in which other type of nuclear-localized proteins are removed by high concentration salt solution (Supplementary Fig. 1)[33]. We identified constituent of MAR proteins by liquid chromatography-mass spectrometry (LC-MS) (Fig. 2a, *SI Appendix*, Supplementary Data 2). We thereby identified 1059 nuclear matrix proteins, including the conserved nuclear matrix proteins SUN (Sad1/UNC-84) proteins AtSUN1 and AtSUN2[34,35] (Fig. 2b). Moreover, we detected several proteins related to chromatin and transcription (*SI Appendix*, Supplementary Data 2), supporting a tight association

**Table 1 | Tandem affinity purification-Mass spectrometric analysis of AHL22**

| Protein | Annotation | % coverage (rep 1) | No. of protein sequences (rep 1) | Protein Score (rep 1) | % coverage (rep 2) | No. of protein sequences (rep 2) | Protein Score (rep 2) |
|---|---|---|---|---|---|---|---|
| AHL22 | AT2G45430 | 12 | 3 | 241 | 21.5 | 5 | 298 |
| AHL15 | AT3G55560 | 49 | 10 | 934 | 52.3 | 11 | 1223 |
| AHL17 | AT5G49700 | 44.2 | 6 | 476 | 20.7 | 4 | 331 |
| AHL13 | AT4G17950 | 18.7 | 5 | 358 | 21.2 | 5 | 305 |
| AHL19 | AT3G04570 | 25.1 | 4 | 306 | 21 | 4 | 255 |
| SPFH | AT5G51570 | 27.7 | 5 | 231 | 19.5 | 4 | 273 |
| AHL27 | AT1G20900 | 19.6 | 3 | 221 | 19.6 | 3 | 197 |
| AHL3 | AT4G25320 | 15.3 | 4 | 215 | 16.3 | 4 | 297 |
| AHL1 | AT4G12080 | 29.2 | 4 | 199 | 39.3 | 8 | 425 |
| AHL4 | AT5G51590 | 14.8 | 3 | 121 | 6.9 | 2 | 90 |
| HTA7 | AT5G27670 | 34.7 | 2 | 119 | 42.7 | 3 | 151 |
| FRS12 | AT5G18960 | 9.1 | 3 | 98 | 4.9 | 2 | 78 |
| VRN1 | AT3G18990 | 8.8 | 3 | 93 | 8.5 | 2 | 113 |
| AHL29 | AT1G76500 | 10.6 | 2 | 88 | 20.5 | 4 | 147 |

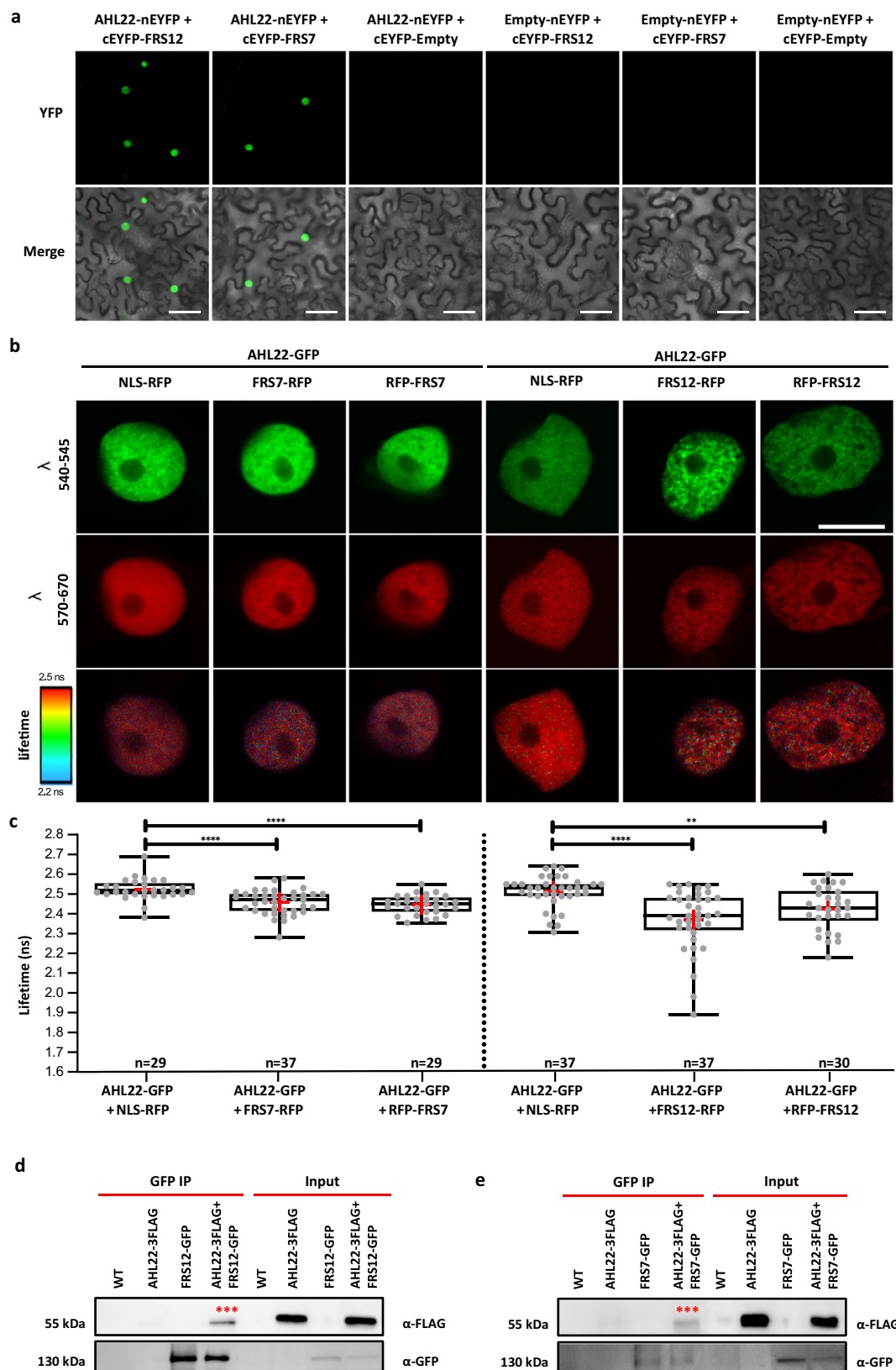

between the nuclear matrix and gene expression. Satisfyingly, we also detected multiple AHL and FRS proteins among the nuclear matrix proteins (Fig. 2b). AHL22, FRS7 and FRS12 were all among the list of proteins identified (Fig. 2b). Hence, we conclude that FRS7 and FRS12 are likely nuclear matrix proteins that can interact with AHL22 at the nuclear matrix.

## AHL22 cooperates with FRS7 and FRS12 to repress hypocotyl elongation

FRS7, FRS12, and other AHL proteins participate in the regulation of hypocotyl elongation[25–28,31], prompting us to ask if they did so cooperatively. We therefore generated an *ahl22 frs7 frs12* triple mutant. The triple mutant showed a significantly longer hypocotyl than the

**Fig. 1 | AHL22 interacts with FRS7 and FRS12. a** Bimolecular fluorescence complementation (BiFC) analysis of AHL22 and FRS7 or FRS12 interaction *in planta*. AHL22 and FRS7 or FRS12 fused with pSITE-nEYFP-N1 and pSITE-cEYFP-C1 vectors, respectively, were cotransformed into *N. benthamiana*. *n* = 3 independent experiments. Scale bar = 50 μm. **b** FRET-FLIM analysis of AHL22 and FRS7 or FRS12 interaction *in planta*. From top to bottom, representative images of nuclei, expresseing various GFP and/or RFP fused proteins in epidermal *N. benthamiana* cells, captured between wavelengths 540-545 nm and between 570-670 nm as well as the corresponding donor fluorescence lifetime calibrated to the adjacent color bar. The respective construct combinations are indicated. Scale bar = 10 um. **c** GFP donor fluorescence lifetime quantifications of the indicated construct combinations presented in a min to max box and whiskers plot showing the median (line in the middle of the box), mean (+ in the box), min and max (whiskers), individual points (gray circle) (*n* > =18). Statistical significance was calculated using a Welch's Anova test combined with Dunnett's T3 multiple comparisons test. P-values are represented as ****(*p* < 0.0001), ***(*p* < 0.001), **(*p* < 0.01). The experiment was performed in two independent infiltrations and the exact number of measurements is indicated per construct combination. **d, e** Co-immunoprecipitation assays showing interactions between AHL22 and FRS12 or FRS7. AHL22-3FLAG and FRS12-GFP or FRS7-GFP were expressed in *N. benthamiana*. WT, empty leaves. IP, immunoprecipitation. Asterisks indicate targeted proteins. *n* = 1 independent experiment.

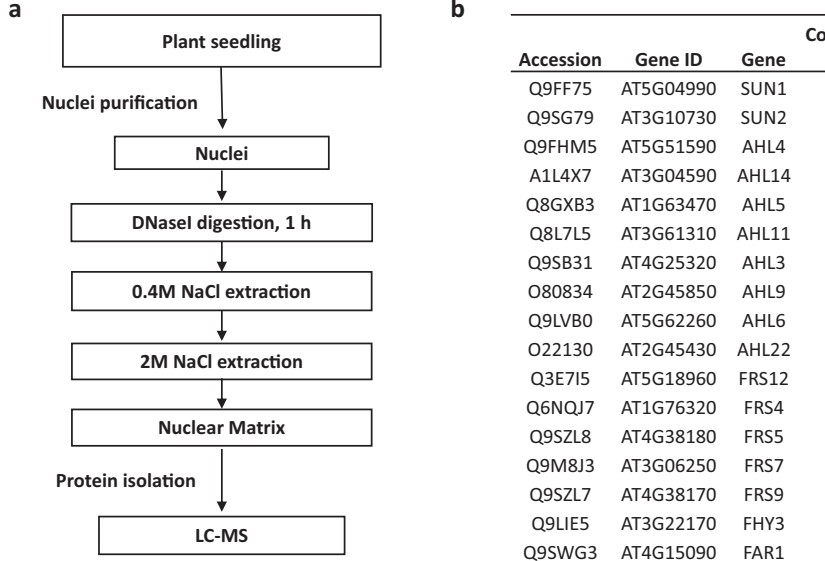

**a**

Plant seedling
↓ Nuclei purification
Nuclei
↓
DNaseI digestion, 1 h
↓
0.4M NaCl extraction
↓
2M NaCl extraction
↓
Nuclear Matrix
↓ Protein isolation
LC-MS

**b**

| Accession | Gene ID | Gene | Coverage [%] | # Peptides |
|---|---|---|---|---|
| Q9FF75 | AT5G04990 | SUN1 | 62 | 24 |
| Q9SG79 | AT3G10730 | SUN2 | 44 | 15 |
| Q9FHM5 | AT5G51590 | AHL4 | 22 | 5 |
| A1L4X7 | AT3G04590 | AHL14 | 20 | 6 |
| Q8GXB3 | AT1G63470 | AHL5 | 9 | 2 |
| Q8L7L5 | AT3G61310 | AHL11 | 11 | 3 |
| Q9SB31 | AT4G25320 | AHL3 | 15 | 4 |
| O80834 | AT2G45850 | AHL9 | 16 | 3 |
| Q9LVB0 | AT5G62260 | AHL6 | 3 | 1 |
| O22130 | AT2G45430 | AHL22 | 9 | 2 |
| Q3E7I5 | AT5G18960 | FRS12 | 36 | 23 |
| Q6NQJ7 | AT1G76320 | FRS4 | 18 | 12 |
| Q9SZL8 | AT4G38180 | FRS5 | 11 | 6 |
| Q9M8J3 | AT3G06250 | FRS7 | 29 | 17 |
| Q9SZL7 | AT4G38170 | FRS9 | 19 | 8 |
| Q9LIE5 | AT3G22170 | FHY3 | 19 | 13 |
| Q9SWG3 | AT4G15090 | FAR1 | 16 | 8 |

**Fig. 2 | Composition of the nuclear matrix-associated proteins. a** Flow diagram representing isolation of nuclear matrix-associated proteins. **b** Identified AHL and FRS-related proteins from the LC-MS analysis, based on three biological experiments, and each experiment includes two technical replicates.

wild-type Col-0 (WT), the *ahl22* single mutant, or the *frs7 fr12* double mutant under LD conditions (16 h 22 μmol·m$^{-2}$·s$^{-1}$ continuous white light, 8 h dark) (*p*-value < 0.01, Fig. 3a, b). In agreement with previous results[27,28], the dominant-negative mutant allele of *AHL29*, *sob3-6*, presented significantly elongated hypocotyl under the same conditions (*p*-value < 0.01, Supplementary Fig. 2a, b). In contrast to the *ahl22 frs7 frs12* triple mutant, transgenic lines individually overexpressing *AHL22*, *FRS7*, or *FRS12* all exhibited significantly shorter hypocotyls than in Col-0 (*p*-value < 0.01, Supplementary Fig. 2c, d). These results indicated that AHL22, FRS7, and FRS12 are likely to coordinately repress hypocotyl elongation.

Given that AHL22, FRS7, and FRS12 are all transcriptional regulators[27,31], we explored how they might control hypocotyl elongation via regulating gene expression. To this end, we generated transcriptome profiles for Col-0, *ahl22*, *frs7 frs12*, and *ahl22 frs7 frs12* seedlings. We identified 3155, 3108, and 3517 differentially expressed genes (DEGs, Log$_2$FC > 1 or Log$_2$FC < −1, with a false discovery rate [FDR] <0.05) in *ahl22*, *frs7 frs12*, and *ahl22 frs7 frs12*, respectively, compared with Col-0 (Supplementary Data 3). We detected substantial overlaps between the DEGs of *ahl22*, *frs7 frs12*, and *ahl22 frs7 frs12* plants (Fig. 3c, and Supplementary Fig. 3a). Notably, the downregulated genes (DGs) and up-regulated genes (UGs) in *ahl22*, *frs7 frs12*, and *ahl22 frs7 frs12* were enriched in similar pathways, such as "response to auxin" and "cytoplasmic translation" for UGs, and "hydrogen peroxide catabolic process" and "response to oxidative stress" for DGs (Supplementary Fig. 3b, c). Among the DEGs, genes responsive to auxin, such as *SAURs*, showed higher expression levels in *ahl22 frs7 frs12* than in *ahl22* or *frs7 frs12* seedlings (Fig. 3d), consistent

with the notion that AHL27 and AHL29 suppress hypocotyl elongation by repressing genes related to auxin signaling pathways[27,28,36]. In line with higher expression of *SAURs* in *ahl22 frs7 frs12*, we found that inhibiting auxin transportation by Naphthylphthalamic acid (NPA) could substantially suppress the long hypocotyl phenotype in *ahl22 frs7 frs12* (Fig. 3e, f), suggesting that AHL22 and FRS7/12 indeed suppress hypocotyl elongation by repressing the auxin pathway.

To further determine the direct function of AHL and FRS proteins in gene expression, especially auxin-related genes, we explored the binding site of the above proteins. We were failed to identify peaks from the ChIP-seq experiment of AHL22-GFP. However, we found an available ChIP-seq dataset of AHL29/SOB3 in a previous publication[37]. AHL29/SOB3 is co-precipitated with AHL22 in the IP-MS experiment (Table 1) and together regulate hypoctyl elongation with AHL22[25]. Therefore, we used the published dataset of AHL29/SOB3 ChIP-seq[37] as the representative binding site of the AHL proteins in hypocotyl regulation, along with the published dataset of FRS12[31]. Consistent with the function of SOB3 and FRS12 as transcription factors, their genome-wide distribution shared a similar pattern, with a preference for promoter-TSS regions (around 50% of all peaks) (Supplementary Fig. 4a). We compared 7,427 SOB3-bound genes[37] to 4,624 FRS12-bound genes[31], of which about 2,000 genes were shared (*p*-value = 6.4e$^{-294}$; Supplementary Fig. 4b, Supplementary Data 4). These ~2000 genes were enriched in phytohormone pathways, including "response to auxin" and "auxin-activated signaling pathway" (Supplementary Fig. 4c), supporting the idea that AHLs and FRS7/12 proteins function as a complex to directly regulate a similar set of downstream target genes in hypocotyl elongation.

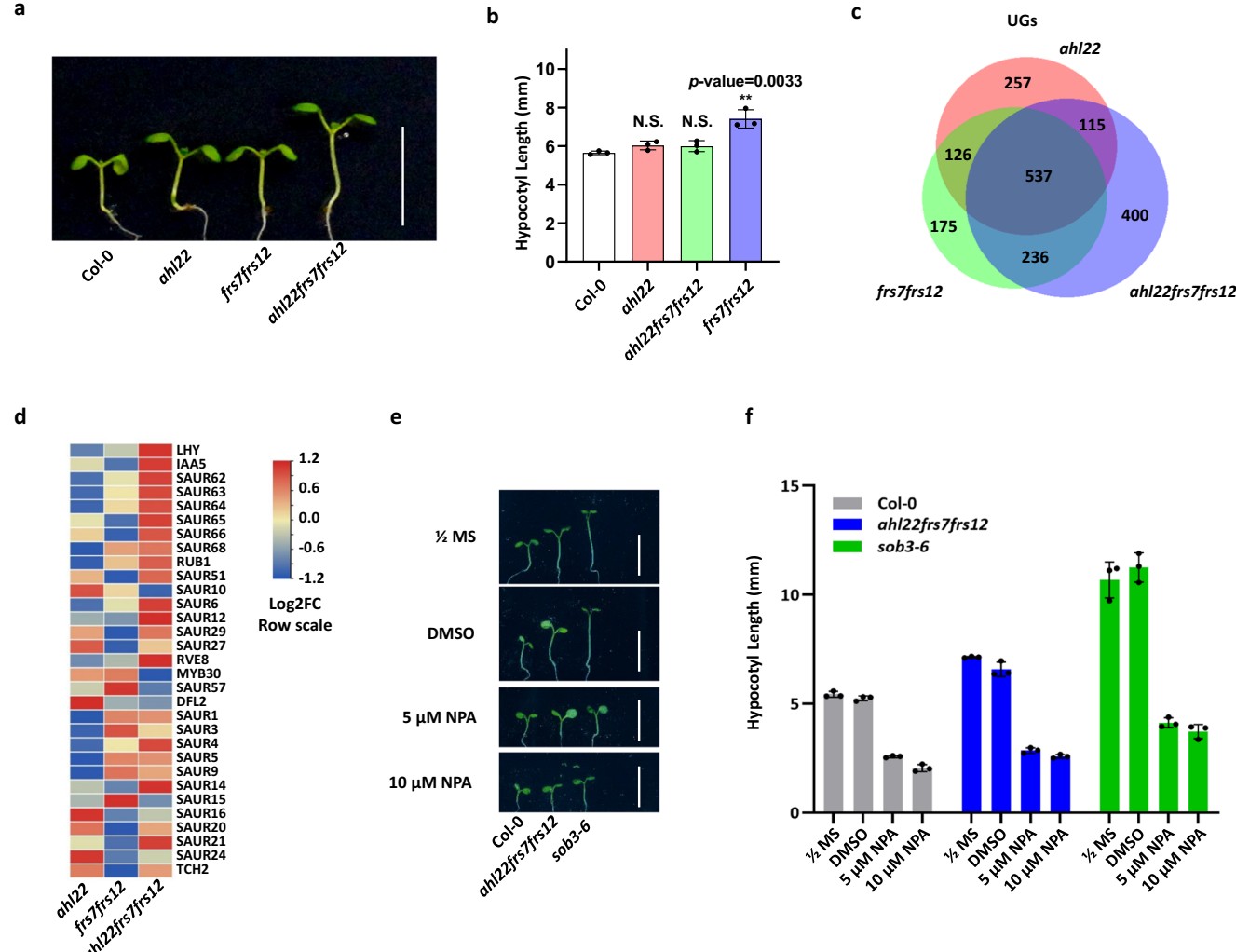

**Fig. 3 | AHL22 and FRS7/FRS12 co-repress hypocotyl growth. a** The hypocotyl phenotype of indicated lines grown vertically under LD conditions (16 h at 23 °C under 22 μmol·m⁻²·s⁻¹ continuous white light, 8 h at 23 °C dark) for 5 days. Scale bar= 1 cm. **b** Hypocotyl lengths of indicated lines grown under LD conditions (16 h at 23 °C under 22 μmol·m⁻²·s⁻¹ continuous white light, 8 h at 23 °C dark) for 5 days. The average length of three independent measurements ± standard deviations are shown. Each measurement with $n = 30$ plants. **c** Venn diagrams showing overlap of upregulated genes (UGs) in *ahl22, frs7 frs12* and *ahl22 frs7 frs12* compared with Col-0. **d** Heat map showing the relative mRNA expression levels of genes responsive to auxin in *ahl22, frs7 frs12* and *ahl22 frs7 frs12* Log2FC row scale was used. **e** The

hypocotyl phenotype of indicated lines grown vertically under LD conditions (16 h at 23 °C under 22 μmol·m⁻²·s⁻¹ continuous white light, 8 h at 23 °C dark) for 5 days. Scale bar= 1 cm. **f** Hypocotyl lengths of indicated lines grown under LD conditions (16 h at 23 °C under 22 μmol·m⁻²·s⁻¹ continuous white light, 8 h at 23 °C dark) for 5 days. Naphthylphthalamic acid (NPA) is a key inhibitor of auxin transportation in plants. DMSO is the solvent for the NPA and included as a control. Average length of three independent measurements ± standard deviations are shown. Each measurement with $n = 30$ plants. Unpaired two-tailed Student's t-test was used to determine significance. N.S. *p value* > 0.05, *p value ≤ 0.05, **p value ≤ 0.01, ***p value ≤ = 0.001, ****p value ≤ = 0.0001.

## MARs are preferentially associated with active epigenetic marks and highly expressed genes

As AHL22, FRS7, and FRS12 are likely nuclear matrix-associated proteins, we hypothesized that they might regulate transcription by recruiting chromatin to the nuclear matrix. To test this hypothesis, we first identified all genomic regions attached to the nuclear matrix by extracting the nuclear matrix and any bound DNA, followed by sequencing, called MAR-seq that was used in mammalian research, including DNA digestion with DNaseI and removal of other proteins by high salt solution (Supplementary Fig. 5a)[38,39]. To exclude the possibility of DNaseI digestion resulting in false positive MAR peaks, we initially compared the peaks obtained from DNaseI-digested DNA with the peaks observed after washing the digested DNA with a high salt solution (Supplementary Fig. 5a). We discovered that the occurrence of MAR peaks distinctly differs from the peaks observed in DNaseI-digested chromatin (Fig. 4a). Therefore, these results confirmed that the peaks in MAR-seq is not due to the preference of DNaseI at

chromatin. We identified 6754 MAR peaks in Col-0 from two replicates (Supplementary Data 5, Supplementary Fig. 5b), overrepresented at two locations over protein-coding gene bodies: downstream of the TSS, and at the transcription end site (TES) (Fig. 4b). Differ from protein-coding genes, no obvious peaks were observed over transposable elements (TEs) (Fig. 4b). Together, these results indicate that MARs are preferentially distributed at protein-coding genes.

We next determined if specific epigenetic marks were enriched in MAR-enriched genes by investigating the distribution of different epigenetic marks using published profiles[40]. In general, MAR-enriched genes were highly correlated with active histone marks, including trimethylation at lysine 4 of histone H3 (H3K4me3), H3K36me3, and histone acetylation, with a preference for the 5' end of protein-coding genes (Fig. 4c). Accordingly, MAR-enriched genes were characterized with low levels of silencing marks, such as H3K27me1/3, H3K9me2, and DNA methylation (Fig. 4c). In agreement with the presence of active histone marks, MAR-enriched genes showed significantly higher

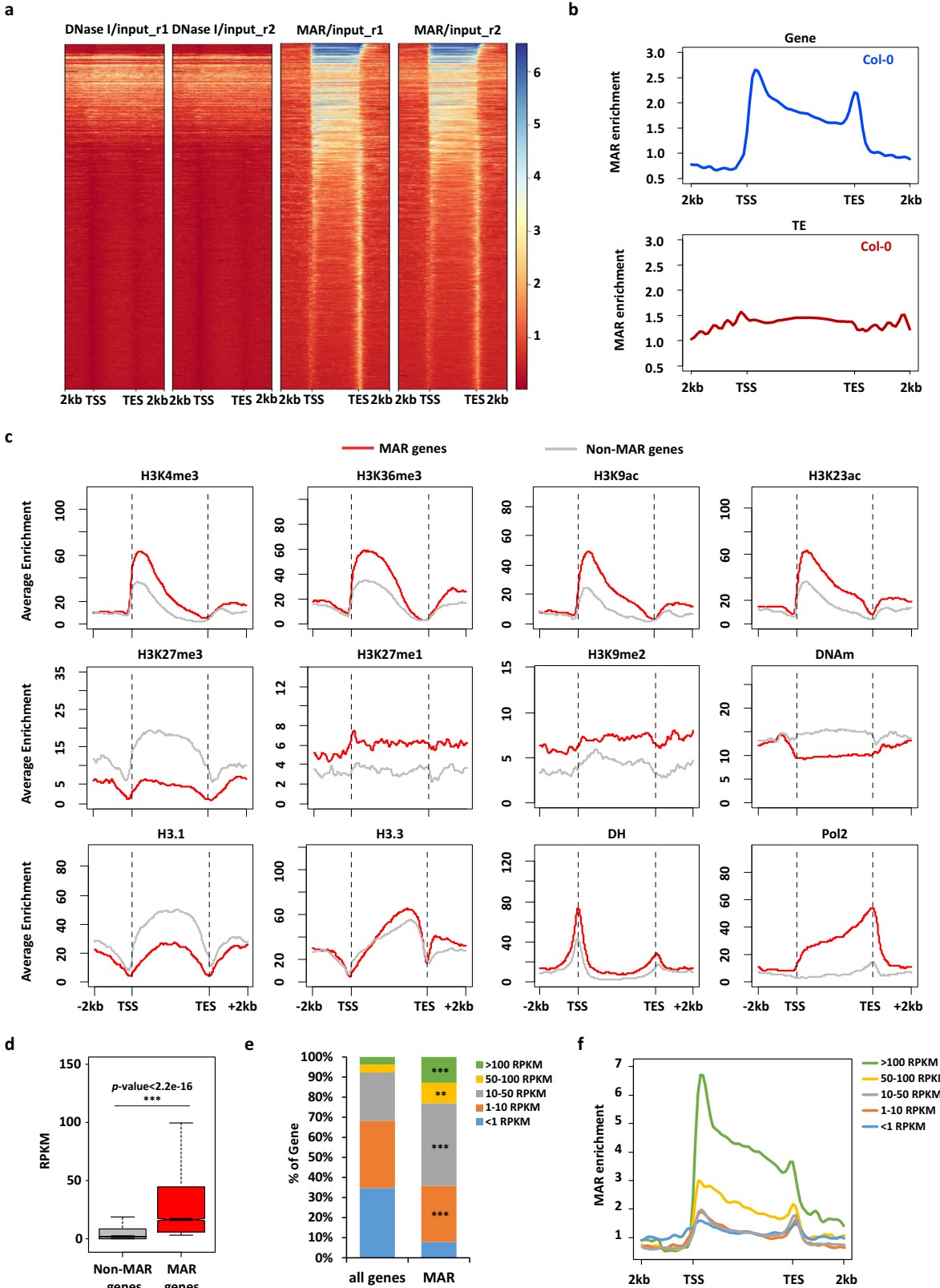

expression levels than other genes (*p*-value < 0.001) (Fig. 4d). A closer look at transcript levels in RNA sequencing indicated that the increased expression levels are driven by more highly transcriptionally active genes (with reads per kilobase of exon per million reads mapped [RPKM] > 10) compared to the genome-wide pattern (Fig. 4e). More-over, highly transcriptionally active genes were more enriched for MARs, especially for genes with RPKM values above 50, relative to the genome-wide average (Fig. 4e). In addition to transcriptionally active genes, some MAR-enriched genes also could be low-expressed genes (RPKM < 1) (Fig. 4f), suggesting that MAR attachment have dual func-tions in transcriptional regulation while it is mainly associated with gene expression.

**Fig. 4 | Characterization of MARs in Col-0. a** Heatmap showing the enrichment of DNaseI/input and MAR/input peaks in genic regions. **b** Metagene plots showing the MAR enrichment of Col-0 over genes (upper) and transposable elements (TEs, down). Genes are scaled to align their TSSs and TESs. Average enrichment means the percentage of regions (calculated from 100-bp windows) enriched for the respective epigenetic mark. TSS: transcription start site, TES: transcription end site. **c** Distribution of average epigenetic features across MARs over genes. Tri-methylation of histone H3 at lysine 4 (H3K4me3), tri-methylation of histone H3 at lysine 36 (H3K36me3), acetylation of histone H3 at lysine 9 (H3K9ac), acetylation of histone H3 at lysine 23 (H3K23ac), tri-methylation of histone H3 at lysine 27 (H3K27me3), mono-methylation of histone H3 at lysine 27 (H3K27me1), di-methylation of histone H3 at lysine 9 (H3K9me2), DNA methylation (DNAm), histone H3.1 (H3.1), histone H3.3 (H3.3), DNase-seq (DH), Pol2 ChIP-seq peaks (Pol2).

**d** Box plot showing the expression level (reads per kilobase of exon per million reads mapped, RPKM) of non-MAR targeted ($n = 21255$) and MAR targeted ($n = 7234$) genes from RNA sequencing, N.S. $p\ value > 0.05$, *$p\ value \le 0.05$, **$p\ value \le 0.01$, ***$p\ value \le 0.001$. (Kolmogorov-Smirnov test). The lower and upper solid lines of the boxplots correspond to the first and third quartiles of the data, the dash lines correspond to each individual data, and the black lines in the middle of the boxes represent the median. The solid lines above the boxplots represent the samples used for the Kolmogorov–Smirnov test. **e** Distribution of genes over expression ranging from less than 1 RPKM, 1-10 RPKM, 10-50 RPKM, 50-100 RPKM and larger than 100 RPKM in MAR targeted genes and all genes. Fisher's exact test, **$P < 0.01$, ***$P < 0.001$. **f** Metagene plot showing the average distribution of MAR enrichment over protein-coding genes grouped by their expression levels (RPKM).

## The AHL22-FRS7-FRS12 complex is involved in MARs attachment

To explore the role of AHL22, FRS7, and FRS12 in MAR attachment, we compared the genome-wide MAR attachment profiles in the *ahl22 frs7 frs12* triple mutant (hereafter referred to as triple mutant) to Col-0. Afterwards, we identified 5576 genes with differential MAR peaks (FDR < 0.05), of which 5457 genes showed a lower MAR enrichment in the triple mutant (Fig. 5a). These results indicated that the AHL22-FRS7/12 complex is involved in MAR attachment. In contrast to genes with differential MAR peaks, many genes showed similar MAR attachment in WT and mutants (examples in Supplementary Fig. 6), suggesting the functional redundancy among different MAR-associated proteins.

Given that MAR-enriched genes are mainly associated with active epigenetic marks and highly transcribed genes (Fig. 4c–e), we hypothesized that lowering MAR attachment in *ahl22 frs7 frs12* at the targeted genes might cause their transcriptional repression. Indeed, genes with fewer MARs had significantly lower expression levels in the triple mutant compared to Col-0 (Fig. 5b, Kolmogorov-Smirnov test, $P < 0.001$). However, when comparing genes with decreased peaks with DGs and UGs, we found both MAR-decreased genes were significantly overlapped with both DGs and UGs (Fig. 5c). Hence, AHL22-FRS7/12-mediated MAR attachment is associated with both gene expression and repression.

Arabidopsis AHL27 and AHL29 inhibit hypocotyl elongation by repressing the expression of genes involved in both auxin and BR signaling pathways, including *YUC8*, *YUC9*, and members of the *SAUR19* subfamily[27,28,36]. Similarly, upregulated genes in the triple mutant with lower MAR peaks were significantly enriched in genes responsive to auxin (Supplementary Fig. 7). In our analyses, several *SAUR* genes showed a decreased MAR enrichment and a higher expression level in the triple mutant compared to Col-0, including *SAUR14/15/16*, *SAUR20/21*, and *SAUR78* (Fig. 5e). We also observed a decline in MAR enrichment in the *sob3-6* mutant (Fig. 5d), further supporting the idea that AHL proteins are involved in MAR attachment to the nuclear matrix. We validated these results by MAR-qPCR at selected *SAUR* genes: relative to Col-0, both the triple mutant and *sob3-6* displayed significantly lower occupancy of MARs over *SAUR* genes, except for *SAUR15*, which might be due to the low overall MAR enrichment over this locus (Fig. 5e, f). Taken together, these data indicate that the AHL22-FRS7-FRS12 complex is responsible for MAR attachment at auxin-response genes, suppressing the expression of multiple *SAUR* genes to inhibit hypocotyl elongation.

## AHL22 recruits HDA15 to repress auxin-responsive genes

We further explored how MARs attaching to the nuclear matrix might influence transcription. Multiple AHLs are known to interact with histone deacetylases, which are essential for transcriptional silencing[32]. Hence, we selected HDA15, whose loss of function also results in longer hypocotyls[41], as a candidate interactor of AHL22. The *hda15-1* mutant displayed a significantly longer hypocotyl phenotype in the same growth conditions as the triple and *sob3-6* mutants (Fig. 6a, b), supporting a role for AHL22 and HDA15 in co-regulating gene expression

to shape hypocotyl elongation. In line with the phenotype of longer hypocotyl, BiFC assays revealed a strong interaction between AHL22 and HDA15, with around 40% of the nuclei showing fluorescence from YFP (Fig. 6c, d). Moreover, FRET-FLIM confirmed the interaction between AHL22 and HDA15 *in planta* (Fig. 6e, f). To further test this idea, we compared the transcriptome profiles of the triple mutant and that of *hda15-1*. We identified 1,179 DEGs (Log$_2$FC > 1 or Log$_2$FC < −1, FDR < 0.05) in *hda15-1* compared with Col-0, with 556 DGs and 623 UGs (Supplementary Data 3). These DGs and UGs showed a significant overlap with the DGs and UGs in the triple mutant, respectively (Fig. 7a), indicating that AHL22-FRS7-FRS12 and HDA15 are likely to act together in transcriptional regulation. Moreover, manual inspection of RNA-seq datasets from the *hda15-1* and *ahl22 frs7 frs12* mutants showed that HDA15 and the AHL22-FRS7-FRS12 complex co-repress a number of genes involved in auxin responses, such as *SAUR* family genes. RT-qPCR analysis confirmed that the expression of *SAUR6*, *SAUR14*, *SAUR15*, *SAUR16*, *SAUR20*, *SAUR21*, *SAUR51*, and *SAUR78* is significantly higher in *hda15-1* compared with the wild type (Fig. 7b), as in the triple mutant, supporting the notion that HDA15 and the AHL22-FRS7-FRS12 complex exert a repressive function for hypocotyl elongation via the auxin signaling pathway.

As a histone deacetylase, HDA15 negatively controls hypocotyl elongation by removing histone acetylation marks[41]. To examine histone acetylation levels, we first performed an immunoblot on total protein extracts from Col-0, the triple mutant, and *hda15-1*, and observed a slight increase of H3 acetylation (H3ac) in the *hda15-1* mutant but not in the triple mutant (Supplementary Fig. 8). We speculate that the AHL22-FRS7-FRS12 complex only influences H3 acetylation marks at MARs, but not across the entire genome, as HDA15 might. We then assessed the H3ac levels by chromatin immunoprecipitation followed by quantitative PCR (ChIP-qPCR) over the promoter regions of selected *SAUR* genes, *SAUR14/15*, *SAUR20*, *SAUR21*, and *SAUR78*. We detected a significant increase of H3ac levels at the promoter regions of these genes in both *hda15-1* and the triple mutant compared to Col-0 (Fig. 7c). In line with increased H3 acetylation levels, the binding of HDA15-GFP at the above SAUR loci was decreased in the triple mutant compared to WT (Fig. 7d). Together, these results support that the AHL22-FRS7-FRS12 complex represses the expression of auxin-response genes by recruiting HDA15 and removing H3 acetylation at targeted loci in hypocotyl growth (Fig. 7e).

## Discussion

The data presented here demonstrate that AHL22, together with FRS7 and FRS12, regulates gene expression modulating hypocotyl elongation in light-grown Arabidopsis seedlings. Multiple AHLs repress hypocotyl growth, such as AHL27 and AHL29. Here, we show that AHL22 interacts with AHL27 and AHL29, suggesting that AHLs usually function as heteromers, in agreement with previous results described for AHL29[25] and AHL10[42]. Moreover, we identified the non-AHL proteins FRS7 and FRS12 as interacting partners of AHL22. The *ahl22 frs7 frs12* triple mutant had a longer hypocotyl relative to the wild type, as also observed in higher-

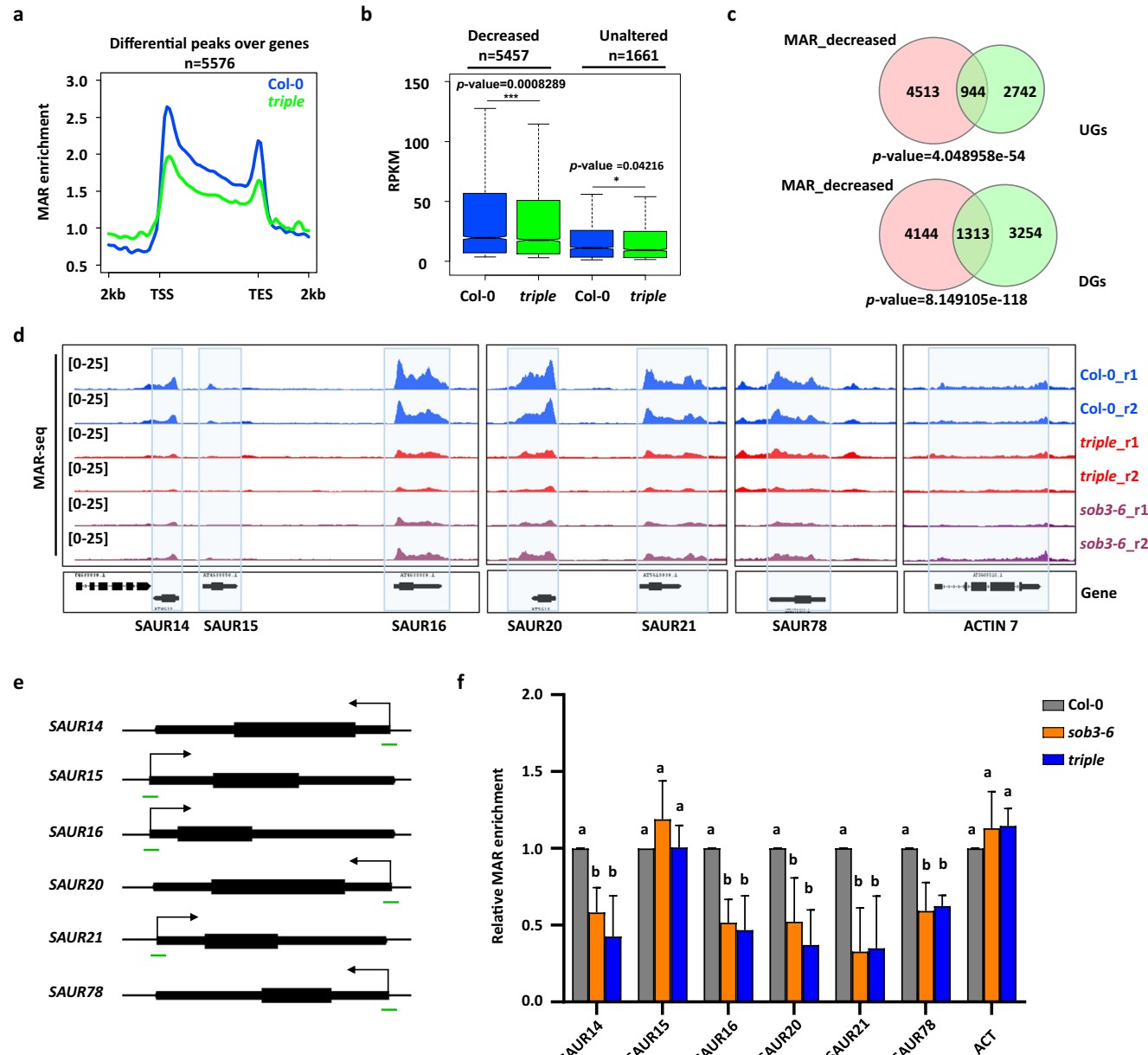

**Fig. 5 | The AHL22-FRS7-FRS12 complex is required for MAR attachment.**
**a** Metagene plot showing the MAR enrichment of all differential peaks (*n* = 5576) in the *triple* mutant compared with Col-0. TSS: transcription start site, TES: transcription end site. **b** Box plot showing the expression level (RPKM) of MAR decreased (*n* = 5,457) and MAR unaltered (*n* = 1,661) genes. N.S. *p value* > 0.05, **p value* ≤ 0.05, ***p value* ≤ 0.01, ****p value* ≤ 0.001. (Kolmogorov-Smirnov test). The lower and upper solid lines of the boxplots correspond to the first and third quartiles of the data, the dash lines correspond to each individual data, and the black lines in the middle of the boxes represent the median. The solid lines above the boxplots represent the samples used for the Kolmogorov−Smirnov test. **c** Venn diagrams showing overlap of MAR decreased genes and UGs (A) or DGs (B)

identified in the *triple* mutant compared with Col-0 (DEGs, Log2FC > 0 or Log2FC < 0, with a false discovery rate [FDR] <0.05). Significance was tested using a hypergeometric test. **d** Genome browser views of two replicates of auxin response genes indicated in blue box in Col-0, *triple* mutant and *sob3-6*. ACTIN 7 was used as control. r1/r2 (replicate1/2). **e** Schematic structures of selected genes. Arrows indicate transcription start sites (TSS). Green lines indicate regions examined by MAR-qPCR. **f** Relative MAR enrichment at selected genes determined by MAR-qPCR in Col-0, *sob3-6* and *triple*. Values are means ± SD of three biological repeats. The significance of differences at each gene was tested using one-way ANOVA with Tukey's test (*P* < 0.05), and different letters indicate statistically significant differences.

order *ahl* mutants. By contrast, individually overexpressing *AHL22*, *FRS7*, or *FRS12* shortened the hypocotyl compared to the wild type, suggesting that AHL22, FRS7, and FRS12 together regulate hypocotyl growth. We also demonstrated that similar to AHL29, the AHL22-FRS7-FRS12 complex regulates hypocotyl elongation by suppressing the expression of a group of *SAUR*s in the auxin signaling pathway. Hence, in addition to multiple AHLs, our results show that another type of MAR-binding protein acts together with AHLs at the nuclear matrix to inhibit hypocotyl growth, namely FRS7 and FRS12.

We characterized the nuclear matrix proteome, which allowed for identification of components of the Arabidopsis nuclear matrix. We identified multiple AHL proteins, supporting the notion that AHL proteins act as MAR-binding proteins in the nuclear matrix. We also identified multiple FRS proteins, including FRS7 and FRS12. The occurrence of other FRSs in the nuclear matrix indicated that FRS family members may commonly function as MAR binding proteins. In addition to AHL, FRS, and other chromatin- or transcription-related proteins, we also noticed multiple RNA binding proteins among the

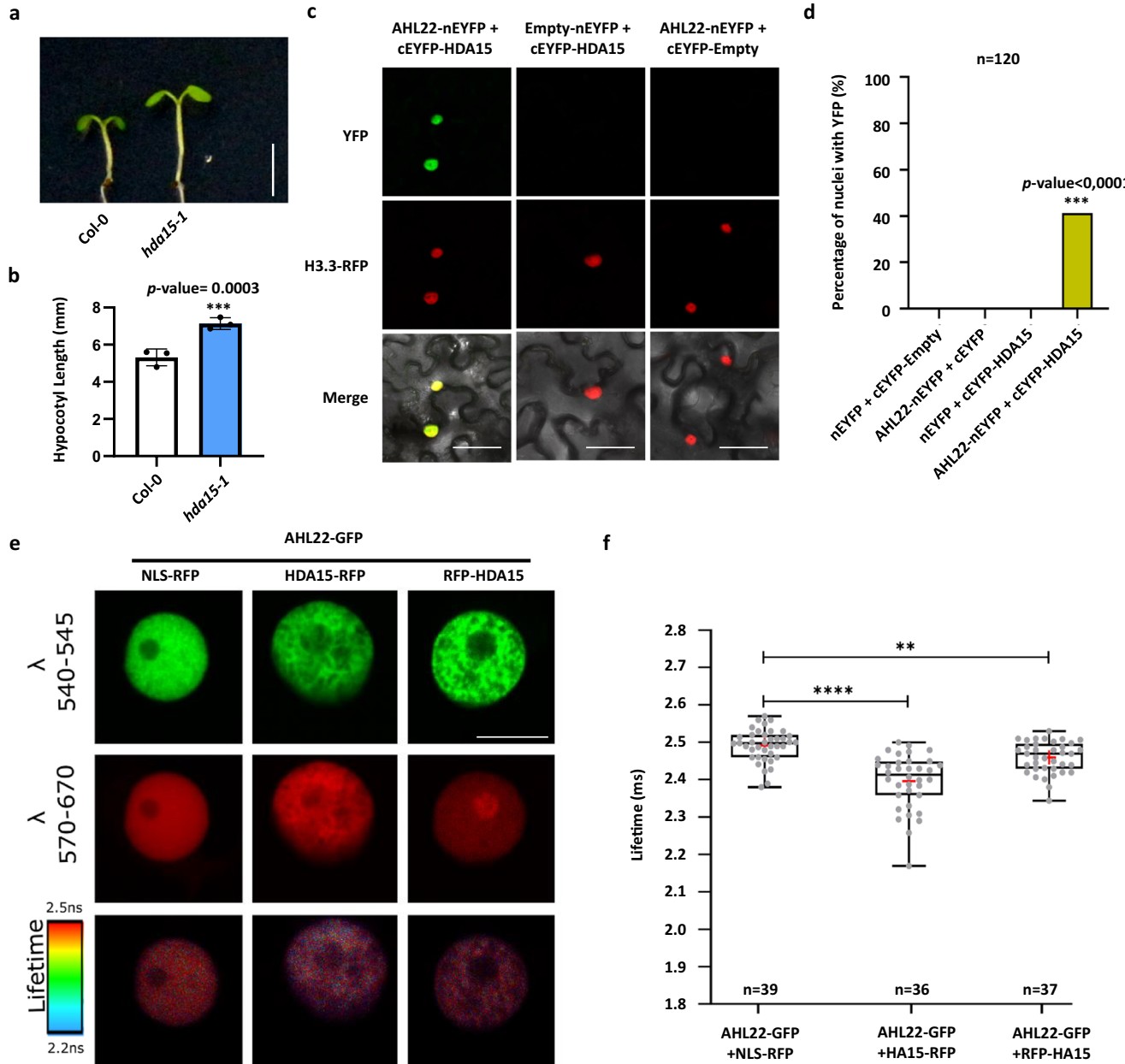

**Fig. 6 | AHL22 cooperates with HDA15 to repress the expression of auxin responsive genes. a** The hypocotyl phenotype of *hda15-1* grown vertically under LD conditions (16 h at 23 °C under 22 µmol·m$^{-2}$·s$^{-1}$ continuous white light, 8 h at 23 °C dark) for 5 days. Scale bar= 0.5 cm. **b** Hypocotyl lengths of Col-0 and *hda15-1*. Average length of three independent measurements ± standard deviations are shown. Each measurement with *n* = 30 plants. Unpaired two-tailed Student's t-test was used. N.S. *p value* > 0.05, *\*p value* ≤ 0.05, *\*\*p value* ≤ 0.01, *\*\*\*p value* ≤ = 0.001, *\*\*\*\*p value* ≤ = 0.0001. **c** Bimolecular fluorescence complementation (BiFC) analysis of AHL22 and HDA15 interaction in vivo. AHL22 and HDA15 fused with pSITE-nEYFP-N1 and pSITE-cEYFP-C1 vectors, respectively, were cotransformed into *N. benthamiana*. H3.3 RFP was used to indicate the nuclei. Scale bar = 50 um. **d** Quantification of the nuclei with both YFP and RFP signal from different pairwise of BiFC experiments. *n* = 120. Fisher's exact test, \*\*P < 0.01, \*\*\*P < 0.001. **e** FRET-

FLIM analysis of AHL22 and HDA15 interaction in vivo. From top to bottom, representative images from nuclear signal, expressed by various GFP and/or RFP fused proteins in epidermal *N. benthamiana* (tobacco) cells, captured between wavelengths 540-545 nm, 570-670 nm and the corresponding donor fluorescence lifetime calibrated to the adjacent color bar. The respective construct combinations are indicated. Scale bar = 10 um. **f** GFP donor fluorescence lifetime quantifications of indicated construct combinations are presented in a min to max box and whiskers plot showing the median (line in the middle of the box), mean (+ in the box), min and max (whiskers), individual points (gray circle) (*n* > =18). Significance was calculated using a Welch's ANOVA test combined with Dunnett's T3 multiple comparisons test. *P*-values are represented as \*\*\*\*(*p* < 0.0001), \*\*\*(*p* < 0.001), \*\*(*p* < 0.01). The experiment was performed in two independent infiltrations and the exact number of measurements was indicated per construct combination.

identified nuclear matrix proteins (Supplementary Fig. 9, Supplementary Data 2), suggesting that the nuclear matrix is also a place of RNA regulation. There is at least one known example of this in mammals. The nuclear matrix protein Matrin3 can bind to RNA transcripts and regulate alternative splicing in Hela cells, besides its role as a MAR-DNA binding protein[43]. Consistent with this example, a type of

extended AT-hook motif was proposed to bind to RNAs[44]. Therefore, multiple lines of evidence highlight the role of the nuclear matrix in RNA regulation in addition to binding to DNA.

Nuclear matrix proteins are associated with transcriptional silencing, such as CHAP (CDP-2/HDA-1-associated protein) from *Neurospora crassa*[45] and AHL10 from Arabidopsis[42]. While H3K9me2 levels also

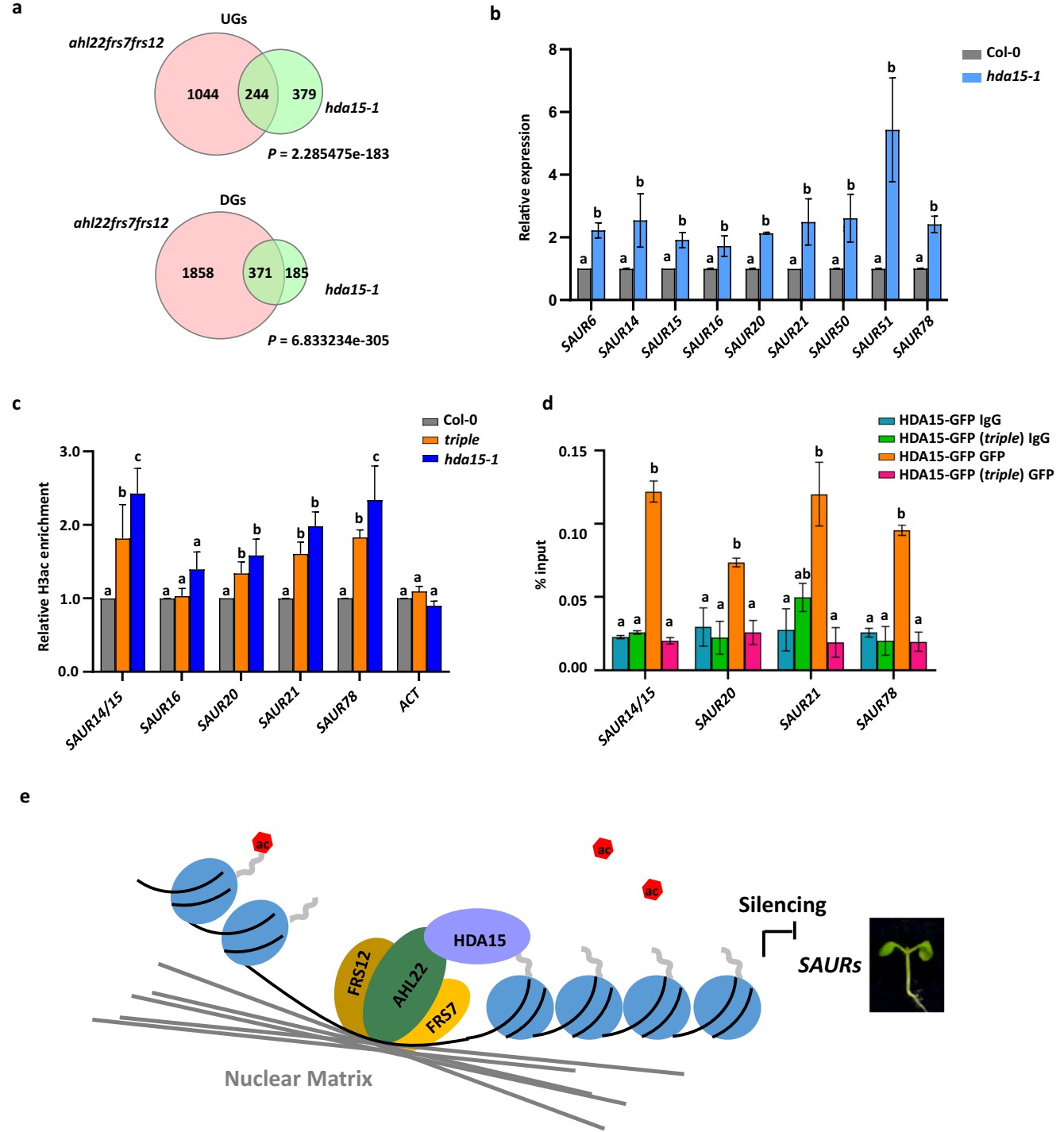

**Fig. 7 | AHL22 recruits HDA15 to remove histone acetylation. a.** Venn diagrams showing overlap of UGs and DGs in *ahl22frs7frs12* and *hda15-1*. Significance was tested using a hypergeometric test. **b** Relative expression level of the selected genes by qRT-PCR from 5-day-old whole plants in *hda15-1*. Values are means ± SD of three biological repeats. The significance of differences at each gene was tested using one-way ANOVA with Tukey's test (*P* < 0.05), and different letters indicate statistically significant differences. **c** Relative H3ac enrichment at selected genes determined by ChIP-qPCR in Col-0, *triple* and *hda15-1* plants. Values are means ± SD of three biological repeats. The significance of differences at each gene was tested using one-way ANOVA with Tukey's test (*P* < 0.05), and different letters indicate

statistically significant differences. **d** IgG and HDA15-GFP enrichment levels at selected genes in indicated lines. The amounts of immunoprecipitated DNA fragments were quantified by qPCR and normalized to input DNA. Values are means ± SD of three biological repeats. The significance of differences at each gene was tested using one-way ANOVA with Tukey's test (*P* < 0.05), and different letters indicate statistically significant differences. **e** Proposed working model. AHL22 interacts with FRS7 and FRS12 at the nuclear matrix and recruits chromatin regions and histone deacetylases, HDA15, to the nuclear matrix and silence the target genes by reducing H3ac level to repress hypocotyl growth.

increased at MAR regions of *FLOWERING LOCUS T* (*FT*) in *AHL22* overexpression lines[32], the *suvh4 suvh5 suvh6* (*su(var)3-9 homolog*) triple mutant did not show any hypocotyl phenotype (Supplementary Fig. 10), suggesting that the function of AHL22 in flowering time determination is independent from H3K9me2-mediated silencing in the context of hypocotyl regulation. We showed that AHL22 interacted with a histone deacetylase, HDA15, whose loss of function resulted in elongated hypocotyl phenotypes, similar as observed for the *ahl22 frs7 frs12* mutant under similar growth conditions. Moreover, we found that AHL22 physically interacted with and recruited HDA15 to the *SAUR* loci to remove H3 acetylation and silence the target genes. Therefore, our results demonstrate that the AHL22-FRS7-FRS12 complex act as a regulatory hub, recruiting both DNA and transcriptional repressors to the nuclear matrix and silence the targeted regions (Fig. 7e). To establish a more direct connection between MAR attachment and hypocotyl growth, it would be interesting to artificially tether the SAUR loci to the nuclear matrix and measure the resulting hypocotyl growth. Intriguingly, our study suggests an implication for the AHL22-FRS7-FRS12 complex in the regulation of multiple SAUR loci. The recruitment of these loci to the nuclear matrix potentially provides a platform for simultaneous epigenetic and transcriptional regulation, enhancing the efficiency of hypocotyl growth regulation. This functionality aligns with the established role of MAR attachment in mammalian systems, contributing to chromatin loop formation[8–10]. Future research endeavors should explore the connection between MAR attachment and chromatin 3D structure in plants. Upon examining the size of SAURs, we discovered that these genes are typically small in size, prompting the question of whether small genes have more MARs. Indeed, genes shorter than 2 kb in length showed a high enrichment of MARs compared to those longer than 2 kb, contributing significantly to the overall decrease in MAR enrichment observed in the triple mutant (Supplementary Fig. 11). Thus, it appears that small genes tend to have more MARs.

While the function of AHL22 in hypocotyl growth depends on transcriptional repression, we determined that MARs are preferentially associated with protein-coding genes but not TEs in the Arabidopsis genome. However, we detected MARs in highly expressed but also lowly expressed genes, a pattern that is similar to the distribution of MARs in other species[46–49]. Therefore, MARs also appear to be associated with gene expression levels. Additional MAR regions in promoters have been shown to enhance the expression of transgenes, also supporting a role for the nuclear matrix in gene expression[50,51]. Apart from MAR regions, the MAR-binding protein AHL29 acts as both a positive and a negative regulator of transcription to regulate petiole growth[37], suggesting that AHLs also act as transcriptional activators. Hence, it will be interesting to determine whether and how nuclear matrix proteins or AHL proteins positively regulate gene expression during plant development.

## Methods

### Plant materials and growth conditions

The *Arabidopsis thaliana* Columbia (Col-0) ecotype was used as the wild type. T-DNA mutants including *ahl22-1* (SALK_018866) and *hda15-1* (SALK_004027) were obtained from Nottingham Arabidopsis Stock Center (NASC). The *sob3-6* mutant was kindly provided by Dr. Michael M. Neff. The *frs7-1 frs12-1* double mutant as well as the overexpression lines (35 S::FRS7-HA and 35 S::FRS12-HA) were kindly provided by Prof. Dr. Alain Goossens[31]. The *ahl22 frs7 frs12* mutant was generated by crossing *ahl22* with *frs7 frs12* and identified by PCR from the F2 population using primers in Supplementary Data 6. Surface-sterilized seeds were sown on ½ MS plates and placed at 4 °C in the dark for 5 days for synchronized germination. Subsequently, the plates were transferred to a growth chamber and grown vertically under LD conditions (16 h at 23 °C under 22 μmol·m⁻²·s⁻¹ continuous white light, 8 h at 23 °C dark with 70% humidity) for 5 days.

### Plasmid construction and generation of transgenic plants

The full-length coding sequences (CDS) with or without stop codon of *AHL22* (AT2G45430), *HDA15* (AT3G18520), *FRS7* (AT3G06250), *FRS12* (AT5G18960), and *H3.3* (AT4G40030), were cloned into TSK108, pDonar201 or pENTR/D (Invitrogen) entry vectors. For AHL22 overexpression lines (35 S::AHL22), the constructs were recombined into a pB7WG2 binary vector using the LR reaction kit (Invitrogen, 11791020). The H3.3-RFP construct was generated by recombining H3.3-pENTR/D into the pUBC-RFP vector. Agrobacterium strain GV3101 was transformed with destination vectors and used for Arabidopsis transformation through the floral dip method[52]. All primers used in this study are listed in Supplementary Data 6.

### Hypocotyl measurement

After transferring the plates to the growth chamber, seeds with synchronized germination after two days were selected for phenotypic analysis. Five-day-old seedlings were scanned, and hypocotyl lengths were quantified using ImageJ software. Three independent measurements were taken, with n = 30 in each measurement. Statistical analyses were performed using an unpaired two-tailed Student's t-test.

### Tandem affinity purification (TAP)

Cloning of AHL22 N-terminal GS^rhino tag[30] fusion under control of the constitutive cauliflower tobacco mosaic virus 35 S promoter and transformation of Arabidopsis cell suspension culture (PSB-D) with direct selection in liquid medium were carried out as previously described[53,54]. TAP experiments were performed in duplicate with 200 mg of total protein extract as input as described in Van Leene et al., 2015. Bound proteins were digested on-bead after a final wash with 500 μL 50 mM NH₄HCO₃ (pH 8.0). Beads were incubated with 1 μg Trypsin/Lys-C in 50 μL 50 mM NH₄OH and incubated at 37 °C for 4 h in a thermomixer at 800 rpm. The digest was separated from the beads, an additional 0.5 μg Trypsin/Lys-C was added, and the digest was further incubated overnight at 37 °C. The digest was centrifuged, and the supernatant was transferred for MS analysis after drying in a Speedvac and storing at −20 °C. Two technical repeats were performed.

### LC−MS/MS Analysis

Peptides were re-dissolved in 50 μl loading solvent A (0.1% TFA in water/ACN (98:2, v/v)) of which 5 μl was injected for LC-MS/MS analysis on an Ultimate 3000 RSLC nano LC (Thermo Fisher Scientific, Bremen, Germany) in-line connected to a Q Exactive mass spectrometer (Thermo Fisher Scientific). The peptides were first loaded on a trapping column (made in-house, 100 μm internal diameter (I.D.) × 20 mm, 5 μm beads C18 Reprosil-HD, Dr. Maisch, Ammerbuch-Entringen, Germany) and after flushing from the trapping column the peptides were separated on an analytical column (made in-house, 75 μm I.D. × 150 mm, 5 μm beads C18 Reprosil-HD, Dr. Maisch). Peptides were eluted by a linear gradient from 98% solvent A' (0.1% formic acid in water) to 55% solvent B' (0.1% formic acid in water/acetonitrile, 20/80 (v/v)) in 30 min at a flow rate of 250 nL/min, followed by a 5 min wash reaching 99% solvent B'. The mass spectrometer was operated in data-dependent, positive ionization mode, automatically switching between MS and MS/MS acquisition for the 5 most abundant peaks in a given MS spectrum. The source voltage was 2.6 kV, and the capillary temperature was 275 °C. One MS1 scan (m/z 400−2000, AGC target 3 × 10⁶ ions, maximum ion injection time 80 ms), acquired at a resolution of 70,000 (at 200 m/z), was followed by up to 5 tandem MS scans (resolution 17,500 at 200 m/z) of the most intense ions fulfilling pre-defined selection criteria (AGC target 5 × 10⁴ ions, maximum ion injection time 80 ms, isolation window 2 m/z, fixed first mass 140 m/z, spectrum data type: centroid, intensity threshold 1.3xE⁴, exclusion of unassigned, 1, 5−8, >8 positively charged precursors, peptide match preferred, exclude isotopes on, dynamic exclusion time 12 s). The HCD collision energy was set to 25% Normalized Collision Energy and the

polydimethylcyclosiloxane background ion at 445.120025 Da was used for internal calibration (lock mass).

## LC−MS/MS Data-analysis and background filtering

From both Thermo raw data files, Mascot Generic Files were created using the Mascot Distiller software (version 2.5.0, Matrix Science). When generating these peak lists, grouping of spectra was allowed with a maximum intermediate retention time of 30 seconds and a maximum intermediate scan count of 5 was used where possible. Grouping was done with 0.005 Da precursor tolerance. A peak list was only generated when the MS/MS spectrum contained more than 10 peaks. There was no de-isotoping and the relative signal-to-noise limit was set to 2. Homology-based protein identification was done using the Mascot search engine (version 2.5.1, Matrix Science) with database TAIRplus (Van Leene et al., 2015, 35,839 sequence entries). Variable modifications were set to methionine oxidation and acetylation of protein N-termini. Fixed modifications were set to carbamidomethylation of cysteines. Mass tolerance on MS was set to 10 ppm (with Mascot's C13 option set to 1) and the MS/MS tolerance at 20 mmu. The peptide charge was set to 2+, 3+ and 4+ and the instrument setting was set to ESI-QUAD. Trypsin was set as the protease used, allowing for 2 missed cleavages, and also cleavage was allowed when arginine or lysine is followed by proline. Only high confident peptides, ranked 1 and with scores above the threshold score, set at 99% confidence, were withheld (Supplementary Data 1). Protein identifications were retained, if they were identified in both TAP experiments, by at least two rank1 peptides with a PSM confidence of >99%, of which at least one is unique to the protein.

Background proteins were filtered out by comparing to a list of 760 non-specific frequent binders built from a large dataset of TAP experiments (Van Leene et al., 2015). However, semi-quantitative analysis based on normalized spectral abundance factors (NSAF), was used in order to allow retention of low abundant frequent binders in the final interactor list (Van Leene et al., 2015). Thereto, for the low abundant frequent binders, identified in 3 to 10 different baitgroups, a Δ NSAF was calculated in the AHL22 TAPs vs large dataset. A threshold of Δ NSAF ≥ 1 was used to retain low abundant frequent binders in the final AHL22 interactor list.

## Bimolecular fluorescence complementation (BiFC)

Full-length coding sequences of AHL22, FRS7, FRS12, and HDA15 were cloned into the pSITE-nEYFP-N1 or pSITE-cEYFP-C1 vectors with the cauliflower mosaic virus (CaMV 35 S)promoter[55] using the LR reaction kit (Invitrogen). Constructs were transiently expressed in 1-month-old *N. benthamiana* plants through Agrobacterium-mediated transformation using an infiltration buffer composed of 10 mM $MgCl_2$, 10 mM MES, and 100 mM acetosyringone with $OD_{600} = 0.8$. Additionally, an Agrobacterium strain expressing Hcpro was added to enhance protein expression. Two days after infiltration, the YFP signal was observed using a Zeiss confocal laser scanning microscope (LSM780). H3.3-RFP served as the nuclear marker for quantitative analysis. The experiment was performed in two independent infiltrations.

## Forster resonance energy transfer measured by Fluorescence Lifetime Imaging Microscopy (FRET-FLIM)

The donor fluorescence lifetime (FLIM) was determined by time-correlated single-photon counting (TCSPC) in *N. benthamiana* (tobacco) epidermal cells transiently expressing the proteins of interest fused to either eGFP (donor) or mRFP (acceptor). Images were collected using an LSM Upgrade Kit (PicoQuant) attached to a Fluo-view FV-1000 (Olympus) confocal microscope equipped with a Super Apochromat 60x UPLSAPO water immersion objective (NA 1.2) and a TCSPC module (Timeharp 200; PicoQuant). A pulsed picosecond diode laser (PDL 800-B; PicoQuant) with an output wavelength of 440 nm at a repetition rate of 40 MHz was used for donor fluorescence

excitation. A dichroic mirror DM 458/515 and a band pass filter BD520/32-25 were used to detect the emitted photons using a Single Photon Avalanche Photodiode (SPAD; PicoQuant). Laser power was adjusted to avoid average photon counting rates exceeding 10000 photons/s to prevent pulse pile up. Image acquisition was done at zoom 10 with an image size of 512 × 512 pixels corresponding to 0.041 um/pixel. A dwell time of 8 us/pixel was used for image acquisition. Samples were scanned continuously (maximum for 1 min) to obtain appropriate photon numbers (around 10000 peak counts) for reliable statistics for the fluorescence decay. Fluorescence life times were calculated using the SymPhoTime software package (v5.2.4.0; PicoQuant). Selected areas of the images corresponding to single nuclei ($n \geq 18$ cells) were fitted by a bi-exponential reconvolution fitting. The calculated IRF (Instrument response function) was included, the background and shift IRF were fixed, and the amplitudes of the lifetimes were set to be positive. The average intensity lifetimes τ for a series of measurements were subjected to the ROUT method (robust regression followed by outlier identificiation) (with Q set to 1%) to remove outliers (PMID: 16526949). The average intensity life times that were maintained after the outlier analysis are presented in a min to max box and whiskers plot showing the median (line in the middle of the box), mean (+ in the box), min and max (whiskers) and individual points (gray circles). Statistical significance was calculated using a Welch's Anova test combined with Dunnett's T3 multiple comparisons test in GraphPad Prism 9.0.0. The experiments were performed in two independent infiltrations.

## Co-immunoprecipitation (Co-IP)

To express tag-fused proteins in *N. benthamiana*, the entry vector TSK108-AHL22 fused with 3FLAG and TSK108-FRS7 or TSK108-FRS12 were recombined into pB7WG2 and pH7FWG2 binary vectors with the 35 S promoter using the LR reaction kit (Invitrogen, 11791020). Agrobacterium strains containing the constructs were co-infiltrated into *N. benthamiana* plant leaves. After 2–3 days, *N. benthamiana* leaves were collected. WT indicates *N. benthamiana* leaves without constructs. Co-IPs were performed as previously described[56]. Briefly, 1-2 g tissues were ground in liquid nitrogen and resuspended in 5 ml of IP buffer (50 mM Tris-HCl pH 7.5, 150 mM NaCl, 5 mM $MgCl_2$, 0. 1% NP 40, 0.5 mM DTT, 10% glycerol, 1 mM PMSF, 1 μg/μL pepstatin, and proteinase inhibitor cocktail (Roche, 11836145001). The mixture was incubated on ice for 20 min and centrifuged at 4000 g for 10 min at 4 °C. The supernatant was filtered through a double layer of Miracloth. The flow-through was incubated with 25 μl GFP-Trap Magnetic Agarose (Chromotek, gtma-20) overnight at 4 °C. After incubation, beads were collected and washed five times with IP buffer, 5 min each wash at 4 °C with rotation. Elution was performed by incubating beads in PBS and 4xSDS buffer at 95 °C for 10 min. Western blotting was performed with an anti-GFP (B-2) antibody (SANTA CRUZ sc-9996, 1:50) and anti-FLAG antibody (F1804, Sigma, 1:1000). Signals were detected with the ChemiDoc (BIO-RAD).

## RNA extraction and RT-qPCR analysis

Total RNA from 5-day-old whole plants was extracted using the Mag-MAX™ Plant RNA Isolation kit (Thermo Fisher #A33784) according to the manufacturer's instructions. Second-strand cDNA was synthesized using the cDNA Synthesis kit (Thermo Fisher, K1612). RT-qPCR reactions contained Solis BioDyne-5x Hot FIREPol®EvaGreen®qPCR Supermix (ROX, Solis BioDyne, 08-36-00008), and the runs were performed in a QuantStudio 6 Flex Real-Time PCR System (Applied Biosystems). Three biological replicates were performed, using GADPH as the reference gene. The primers used are listed in Supplementary Data 6.

## RNA-seq

Total RNA from 5-day-old whole plants was extracted using the RNeasy Plant Mini Kit (Qiagen, 74904). For RNA-seq, libraries were constructed with the VAHTS mRNA-seq V6 Library Prep Kit (Vazyme, NR604-01)

following the manufacturer's instructions. The libraries were sequenced at Novogene (UK) using a Novaseq instrument in 150-bp paired-end mode. Two or three biological replicates were performed.

## Isolation of the nuclear matrix

The nuclear matrix was isolated as follows. Briefly, 5-day-old whole Arabidopsis plants grown under LD condition (16 h at 23 °C under 22 μmol·m$^{-2}$·s$^{-1}$ continuous white light, 8 h at 23 °C dark with 70% humidity) were collected and chopped in 1 ml LB01 buffer (2 mM Na$_2$EDTA, 20 mM NaCl, 2 mM EDTA, 80 mM KCl, 0.5 mM spermine, 15 mM β-mercaptoethanol, 0.1% Triton X-100, pH 7.5)[38]. Another 9 ml LB01 buffer was used to wash the nuclei into a 1-layer miracloth filter followed by a one-time filter through 30 μm CellTrics (Sysmex, Germany, 04-0042-2316). The nuclei mixture was centrifuged at 1500 g for 10 min at 4 °C. The nuclei pellet was further washed with DNase I buffer (20 mM Tris-HCl pH 7.4, 20 mM KCl, 70 mM NaCl, 10 mM MgCl$_2$, 0.125 mM spermidine 1 mM PMSF, 0.5% Triton-X 100) until it turned white. After centrifugation, the nuclei were resuspended in 200 μl DNase I buffer, and an aliquot of nuclei was stored for isolation and estimation of total genomic DNA as control. DNase I (Thermo Scientific, EN0525) was added and incubated at 4 °C for 1 h. Nuclei were collected by centrifugation at 3000 g for 10 min at 4 °C. Digestion was followed by extraction with 0.4 M NaCl for 5 min twice on ice in Extraction Buffer (10 mM Hepes pH 7.5, 4 mM EDTA, 0.25 mM spermidine, 0.1 mM PMSF, 0.5% (v/v) Triton X-100). Followed by another two rounds of extraction with 2 M NaCl for 5 min on ice in Extraction Buffer. The final nuclear matrix pellet was washed twice with Wash Buffer (5 mM Tris-HCl pH 7.4, 20 mM KCl, 1 mM EDTA, 0.25 mM spermidine, 0.1 mM PMSF).

The nuclei from different steps as indicated were checked by 4′,6-diamidino-2-phenylindole (DAPI) staining and analyzed using an LSM780 confocal microscope (Zeiss) to validate DNA content in the nuclei (Supplementary Fig. 1). For protein validation, after chopping, the nuclei mixture was separated into several same-volume nuclei and followed the extraction procedure. Proteins were extracted from the supernatant or pellet as indicated. All proteins were separated on 10% Tricine-SDS-PAGE gels[57] followed by Coomassie staining. The gel was visualized with the ChemiDoc (BIO-RAD) (Supplementary Fig. 1).

## Proteome analysis of the nuclear matrix

Proteins present in the nuclear matrix, isolated from three independent preparations of 1 g of plants as described above, were solubilized in RapiGest SF (Waters, Germany) in 50 mM ammonium bicarbonate for 1 h at 37 °C under shaking conditions, then reduced in a final concentration of 2.5 mM dithiothreitol for 10 min at 60 °C, and alkylated in a final concentration of 7.5 mM iodoacetamide for 30 min at room temperature. For protein digestion, trypsin was added in a 1:50 (weight to weight) ratio and incubated overnight at 37 °C[58]. Desalting of peptides was done using Peptide Desalting Spin Columns (Pierce, Thermo Scientific, United States) following the manufacturer's instructions. Peptides were resuspended in 2% acetonitrile/0.1% trifluoroacetic acid to a volume of 30 μl and 2 μL of protein digest was analyzed using nanoflow liquid chromatography on a Dionex UltiMate 3000 system (Thermo Scientific) coupled to a Q Exactive Plus mass spectrometer (Thermo Scientific)[59]. Peptides were loaded onto a C18 trap column (0.3 × 5 mM, PepMap100 C18, five μm, Thermo Scientific) and then eluted onto an Acclaim PepMap 100 C18 column (0.075 × 250 mM, 2 μm particle size, 100 Å pore size, Thermo Scientific) at a flow rate of 300 nl min$^{-1}$. The mobile phases consisted of 0.1% formic acid (solvent A) and 0.1% formic acid in 80% ACN (solvent B). Peptides were separated chromatographically by a 200 min gradient from 2% to 44% solvent B, with the column temperature set at 40 °C. A Nanospray Flex ion source was used for electrospray ionization of peptides, with the spray voltage set at 1.80 kV, capillary temperature at 275 °C, and the S-lens RF level at 60. Mass spectra were acquired in positive ion and

data-dependent mode. Full-scan spectra (375 to 1,500 m/z) were acquired at 140,000 resolution, and MS/MS scans (200 to 2,000 m/z) were conducted at 17,500 resolution. The maximum ion injection time was 50 ms for both scan types. The 20 most intense MS ions were selected for collision-induced dissociation fragmentation. Singly charged ions and unassigned charge states were rejected; the dynamic exclusion duration was set to 45 s. Each sample was measured in duplicate. The raw files were processed using Proteome Discoverer 2.4 and Sequest HT engine (Thermo Scientific), searching the SwissProt *Arabidopsis thaliana* (TaxID = 3702) database (downloaded July 2021). Precursor ion mass tolerance was set to 10 ppm and fragment ion mass tolerance was set to 0.02 Da. False discovery rate (FDR) target values for the decoy database search of peptides and proteins were set to 0.01 (strict level for highly confident identifications). Further parameters for database search were: peptide tolerance: ten ppm; fragment ion tolerance: 0.02 Da; tryptic cleavage with max. two missed cleavages; carbamidomethylation of cysteine as a fixed modification and oxidation of methionine as a variable modification; minimum peptide length: 6. The result lists were filtered for high confident peptides, and their signals were mapped across all LC-MS experiments and normalized to the total peptide amount. Protein abundance quantification was done using the Top N average method, (N = 3). The result lists were firstly filtered, and proteins were only kept for further investigation if they fulfilled the following characteristics: identified by at least two peptides or by one peptide representing at least 10% protein coverage. Moreover, we further removed 2872 the proteins annotated as ribosome or other cytoplasmic localized proteins in GO analysis, which were considered as background from the nuclear matrix purification.

## MAR-qPCR and MAR-seq

DNA was extracted from the isolated nuclear matrix and the input sample. To remove RNA, RNase A (Thermo Scientific, EN0531) was added to a final concentration of 20 μg/ml and incubated for 30 min at 37 °C. This was followed by digestion with 100 μg/ml Proteinase K (Ambion, AM2546) at 55 °C for 1 h. DNA was recovered by extraction with phenol:chloroform:isoamyl alcohol (25:24:1) and ethanol precipitation. The isolated input DNA was fragmented using a Bioruptor® Plus sonication device (Diagenode) to obtain the desired DNA fragment size (enriched at 200 bp). The amount of fragmented DNA was quantified by qPCR. Three biological replicates were performed. Statistical analyses were performed by using one-way ANOVA with Tukey's test (P < 0.05), and different letters indicate statistically significant differences. The primers used are listed in Supplementary Data 6.

Libraries were prepared using the NEBNext® Ultra™ II DNA Library Prep Kit (NEB, E7645) according to the manufacturer's instructions. The libraries were sequenced at Novogene (UK) via a Novaseq instrument in 150-bp paired-end mode. Two biological replicates were performed.

## Bioinformatic analysis

For RNA-seq analysis, reads were mapped to the TAIR10 wild-type Arabidopsis genome with HISAT2[60] in paired-end mode. Differentially expressed genes were analyzed via the Subread[61] and DESeq2[62] R packages with a 0.05 false discovery rate (FDR). The expression level of each gene, RPKM, was calculated using a home-made script in R. The heatmap of DEGs were generated using TBtools[63]. GO term enrichment was determined at https://david.ncifcrf.gov/summary.jsp.

For MAR-seq analysis, quality control and adapter trimming of MAR-seq reads were performed with Fastp[64]. Reads were aligned to the TAIR10 reference genome using Bowtie2. Mapped reads were deduplicated using MarkDuplicates (https://broadinstitute.github.io/picard). Coverage was estimated and normalized to 10 million reads. MAR-seq peaks in wild-type and mutants were called by SCIER2[65]. The IGB genome browser was used to visualize the data and to generate screenshots. Boxplots and metagene plots were generated with R.

Kolmogorov-Smirnov tests were performed at https://scistatcalc.blogspot.com. GO term enrichment was determined at https://david.ncifcrf.gov/summary.jsp.

## Histone extraction and western blot

The histone was extracted by the EpiQuik Total Histone Extraction Kit (EPIGENTEK, OP-0006-100) and followed the manufacturer's recommendations, using 5-day-old whole plants. Extracted histones were separated on 10% Tricine-SDS-PAGE gels[57]. Primary antibodies used included anti-H3 (Sigma/H9289), anti-H3ac (Millipore/06-599), and anti-H4ac (Millipore/06-866). Signals were detected with the ChemiDoc (BIO-RAD).

## ChIP-qPCR

Approximately 1 g of 5-day-old whole plant grown under LD conditions (16 h at 23 °C under 22 µmol·m$^{-2}$·s$^{-1}$ continuous white light, 8 h at 23 °C dark with 70% humidity) was collected. The fresh tissue was cross-linked in 1x PBS (0.01% Triton, 1% formaldehyde) and vacuum infiltrated for 10 min on ice[66]. The cross-linking was stopped by adding glycine to a final concentration of 125 mM. Cross-linked tissues were ground to a fine powder in liquid nitrogen and resuspended in 7 ml of Honda buffer (0.4 M Sucrose, 2.5% Ficoll, 5% Dextran T40, 25 mM Tris-HCl, pH 7.4, 10 mM MgCl$_2$, 0.5% Triton X-100, 0.5 mM PMSF, 10 mM β-mercaptoethanol, proteinase inhibitor cocktail (Roche, 11836145001)). The mixture was incubated on ice for 20 min followed by two times filter through Miracloth and one time through 30 µm CellTrics (Sysmex, Germany, 04-0042-2316) and centrifuged at 1500 g for 10 min at 4 °C. The pellet was resuspended in 150 µl Nuclei lysis buffer (50 mM Tris–HCl, pH 8.0, 10 mM EDTA, 1% SDS, 0.1 mM PMSF, 1 µM pepstatin A, Protease Inhibitor Cocktail) and sonicated by using a Bioruptor® Plus sonication device (Diagenode) to obtain the desired DNA fragment size (enriched at 500 bp). The following procedures were described by ref. 66. Immunoprecipitations were performed with anti-H3 (Sigma, H9289), anti-H3ac (Millipore, 06-599), anti-GFP (Invitrogen, A11122) and IgG Isotype Control (Invitrogen, 026102) antibodies. After the de-crosslinking, DNA was purified according to the IPure kit v2 Kit manual (Diagenode, C03010015). The amount of immunoprecipitated DNA was quantified by qPCR. Three biological replicates were performed. Statistical analyses were performed by using one-way ANOVA with Tukey's test ($P < 0.05$), and different letters indicate statistically significant differences. The primers used are listed in Supplementary Data 6.

## Reporting summary

Further information on research design is available in the Nature Portfolio Reporting Summary linked to this article.

## Data availability

All sequencing data have been submitted to the NCBI Gene Expression Omnibus (GEO) under accession number GSE215135. Proteome raw data has been deposited at MassIVE under the dataset ID MSV000090406 [https://massive.ucsd.edu/ProteoSAFe/dataset.jsp?task=aece25b2191440c8ac533865ee58545c]. Source data are provided with this paper.

## Code availability

All scripts used in this study are available upon request.

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

## Acknowledgements

We thank Dr. Michael M. Neff and Prof. Dr. Alain Goossens for kindly sharing seeds. We thank Prof. Chang Liu for the help in bioinformatics

analysis. This research was supported by the intramural funding from IPK to HJ, and a grant from the DFG to HJ (JI 347/6-1).

## Author contributions

L.X., S.Z., K.W., E.V.D.S., G.D.J., A.B., D.V.D., E.M., M.S., D.I., and J.C., executed the experimental procedures. K.W. analyzed the composition of nuclear matrix-associated proteins; E.V.D.S. and G.D.J. performed the IP-MS analysis for AHL22; A.B. and D.I. shared material, E.M. performed FLIM measurements. D.V.D., A.B. and E.M. analyzed data; L.X. and H.J. analyzed the NGS data. L.X. and H.J. performed the experimental design. L.X. and H.J. wrote the manuscript. All authors discussed the results and commented on the manuscript.

## Funding

## Competing interests

The authors declare no competing interests.
