## [Peer Review File · Nature Communications]

Chromatin attachment to the nuclear matrix represses hypocotyl elongation in *Arabidopsis thaliana*Reviewer #1 (Remarks to the Author):

Xu et al. reported detailed analysis of AHL22 along with FRS7 and FRS12 to delineate hypocotyl elongation of the corresponding triple mutant Arabidopsis. The experiments were well conducted and results are very well described and coherent. AHL22, FRS7 and FRS12 are likely to bind to nuclear matrix regions and suppress auxin-related gene expressions and subsequent hypocotyl elongation of seedlings. The authors proposed a model that a group of genes called SMALL AUXIN UP RNAs (SAURs), which contained matrix attachment regions (MARs), were promoted to attach to the nuclear matrix, where the histone deacetylases like HDA15 are recruited, and epigenetically regulated.

The story is very clear, but the key point still needs confirmation whether chromatin attachment to the nuclear matrix really happens first and subsequently it leads to repress hypocotyl elongation. The cause and effect relationship is not clear. There is still possibility that seedlings show hypocotyl elongation, then AHL22/FRS7/FRS12 would interact with each other and epigenetic changes could be observed as an effect. I would like to know how the authors could reply to the question.

To substantiate the model more conclusively:

Is it possible to make artificial/inducible tethering of AHL22/FRS7/FRS12 to matrix attachment regions to see subsequent seedling growth?

Major points:

Line 77: Why did the authors start AHL22 as the key factor in this study? Readers could not understand what is AHL22 and why AHL22 in the first place.

Table 1: Out of these proteins, why did the authors start the study with AHL22, FRS7 and FRS12, but not with AHL27 or AHL29?

Fig.1C: Some differences were observed between AHL22-FRS7 and AHL22-FRS12. Does it not make biological significance? + (mean) marks are hard to see.

Minor comments

Abbreviations UG and DG firstly appeared in lines 127 and 129 without definition. The definition appeared later in line 212.

Fig.4C: Epigenetic features should be respectively defined.

Reviewer #2 (Remarks to the Author):

In this study, Linhao and colleagues investigated the role of chromatin attachment to the nuclear matrix in regulating hypocotyl elongation in Arabidopsis thaliana. They identified a nuclear matrix binding protein, AHL22, that interacts with transcriptional repressors FRS7 and FRS12 to suppress the expression of a group of genes known as SAURs. The repression of SAURs depends on their attachment to the nuclear matrix. The AHL22 complex recruits the histone deacetylase HDA15 to the SAUR loci, leading to the removal of H3 acetylation and the suppression of hypocotyl elongation. This study reveals a novel mechanism by which nuclear matrix attachment to chromatin regulates gene expression and hypocotyl elongation. In principle, this work could be potentially very interesting as it provides (to my knowledge) the first characterization of the nuclear matrix and MARs in plants, while functionally linking its role to gene regulation. These results could have broad significance for research in plants however, I have a few questions would the work as it stands.

1. My primary concern relates to the protocol used for MARs isolation, as this impacts most of the results in the study. The authors used a protocol from Drosophila, but how can we be certain that the composition is the same and that the protocol will work equally well in plants? In figure S1 the

authors show images of nuclei at different steps of the protocol but I find them strange in appearance. For instance, no chromocenters are visible on the DAPI channel (instead, there are some rounded bodies that stain less with Dapi, which are also visible in brightfield images) and the nucleus size is too large (almost 50um in diameter). Also, after DNase I digestion no nuclei should be visible but in FigS1 the input image and after DNaseI digestion look similar. This leads me to question the accuracy of their protocol. Could the authors clarify this and maybe also try their protocol on AHL22-GFP lines, observing loss of DNA but presence of AHL22 after DNaseI digestion?

2. The nuclear matrix was first reported in the 70s but since then numerous studies have failed to provide evidence for its existence and there's still some debate in the field. I think the authors should at least make a reference to that. I also felt that generally more introduction to nuclear matrix and MARs is necessary – for instance what is known in plants and why a nuclear matrix is necessary. In fact, this last point is not very well discussed. It is not clear to me what is the relevance of recruitment to nuclear matrix (especially since no correlation with gene expression or repression was observed)? For instance, could SAURs be recruited at the same nuclear location for co-regulation? I understand this may be beyond the scope of this paper but I felt like more discussion on this is needed.

3. The authors observed that AHL22-FRS7-FRS12 is required for MARs attaching to the nuclear matrix and that FRS proteins may function as MAR-binding proteins. To further support this claim I think it would be important to compare the genome-wide MAR attachment profiles in ahl22 single mutant alone with the triple mutant and Col-0.

4. Why HDA15 is not present on the nuclear matrix proteome or in the TAP-MS analysis of AHL22? It would be helpful if the authors could discuss this.

Minor comments:

- Citation missing: Igor V. Tetko et al., 2006 Plos Computational Biology. Showing evidence for nuclear matrix importance in plants.
- Line 98: salt instead of self.
- Line 464 and 467 Fig.S1 instead of Fig.S14.
- Figure 3: missing label on x-axis of graph (3B) – triple mutant.
- BiFC is also a method to test for protein interaction in planta. However, FRET/FLIM is presented as the only method doing so. Correct this miss reference.

Reviewer #3 (Remarks to the Author):

In this paper "Chromatin attachment to the nuclear matrix represses hypocotyl elongation in *Arabidopsis thaliana*" the authors present evidence that a nuclear matrix binding protein physically interacts with two transcriptional repressors at matrix attachment regions and regulates hypocotyl elongation by recruiting histone deacetylase to suppress the expression of auxin-response genes. I found this to be an intriguing story that ties gene regulation during plant development to nuclear matrix attachment and the concept of proteins exerting their function by acting as scaffolds for others to assemble into complexes.

As I was reading the paper, I noted a few comments/questions on each of the sections.

1) AHL22 interacts with FRS7 and FRS12 – This is demonstrated thoroughly by several methods including tandem affinity purification, followed by BiFC and FRET-FLIM, and co-immunoprecipitation (table 1, figure 1). The methods sections for these experiments in particular could benefit from a proof-reading and some clarification. E.g. the 35S promoter is from cauliflower (not tobacco) mosaic virus, OD=0.8 should probably specify the wavelength, and there

was a reference missing listed simply by its PMID number. I noticed several typos, such as quantitative, combinations, and intensity that would all be easily caught by a spellcheck. The BiFC section mentions that H3.3-RFP was used as a marker protein but does not specify its source. It also mentions that percentage indicates strength of protein interactions, but this isn't included in figure 1. It is included later in figure 6D, which makes me wonder why it was omitted from figure 1.

2) This interaction happens at the nuclear matrix – This is inferred from identifying all three proteins via LS-MS in isolated nuclear matrix fractions (figure 2). Nuclear matrix was isolated by high concentration salt solution (typo salt in line 98 needs fixing) and resulted in several hundred proteins being identified including some conserved nuclear matrix proteins such as SUN1 and SUN2. I checked the data table for some other known nuclear matrix constituents and found them missing (e.g. CRWN1/LINC1). Some matrix proteins are notoriously difficult to solubilize and may not have been detectable by LS-MS. Thus, I think the sentence in the discussion claiming that this “allowed the identification of the composition of the Arabidopsis nuclear matrix” is probably overly optimistic and I would suggest changing that to say it allowed for identification of components of the nuclear matrix. In the proteome analysis section, the authors mention removing cytoplasmic proteins considered background, which makes me wonder how many of these were removed compared to the total. Importantly though all three proteins that were subject of the study cofractionated, corroborating the conclusion that FRS7 and FRS12 are likely nuclear matrix proteins that can interact with AHL22 at the nuclear matrix. In the table in figure 2 listing the AHL and FRS-related proteins identified, I thought the last column showing 3 replicates for everything was not needed and just adding visual clutter as it could be easily mentioned in the legend.

3) AHL22 cooperates with FRS7 and FRS12 to repress hypocotyl elongation – This is demonstrated by the phenotypes of mutant lines with a triple mutant showing significantly longer hypocotyls compared to single or double mutants, suggesting genetic interaction (figure 3). Conversely, overexpressing lines showed shorter hypocotyls. The methods section lists the sources for the single and double mutants, as well as the overexpressing lines, but makes no mention of how the triple mutant was generated, nor is there any experimental evidence provided confirming its triple mutant status. I'm assuming the triple mutant line was generated by the authors using genetic crosses and PCR for genotyping, but that info is missing and should be added.

4) The mode of repression is via regulating gene expression – Evidence for this is provided by comparing transcriptome profiles, presumably generated by RNA-seq listed in the methods (this might need clarification), and finding substantial overlap between differentially expressed genes in the mutants compared to wildtype. Further things I noted as needing clarification are defining UGs and DGs in line 127 (it's done in line 212 but should be done here when those abbreviations are first used) and the mention of a volcano plot in line 505. I did not see any volcano plots being used in either the paper or the supplemental figures.

5) This gene repression specifically targets the auxin pathway – This conclusion is backed up by finding higher expression levels of genes responsive to auxin in the mutant lines, and conversely an inhibitor of auxin transport suppressing the long hypocotyl phenotype in the mutants. That is nice data but I found the choice of graph in figure 3F a little puzzling. The line graph is suggestive of a sequence such as a timeline. I suspect it is meant to connect increasing levels of NPA but I think a bar graph would work better here. I also think the figure legend should define NPA (it's spelled out in the text but not in the figure), and I assume the DMSO is the solvent for the NPA and included as a control.

6) AHLs and FRS7/12 possibly interact directly with the promoters of the target genes – The authors infer this from published datasets for related proteins that co-precipitated with AHL22 because they were not able to get data from the ChIP-seq experiment with AHL22. This is an obvious gap in the data but an argument can be made that these proteins interact as a complex and thus likely target the same genes.

7) MARs are enriched at TSS and TES of protein-coding genes – This is demonstrated by MAR-seq, which is sequencing DNA extracted from isolated nuclear matrix (figure 4). This appears to be a novel method developed in animals applied to plants. TSS should be defined where it is first used

as an abbreviation. Another abbreviation that needs clarification is TEs vs. TES. It's not explained in either the text or the figure legend, but from the context it is clear TE is meant to contrast with protein-coding genes. Transposable elements? Also, the figure legend mentions aligning TTSs, but there is nothing labeled TTS in the figure. It is called a metagene plot in the figure legend but metaplot in the bioinformatics section of the methods.

8) MAR enrichment correlates with active epigenetic marks of highly expressed genes – This is demonstrated with more metagene plots comparing the presence of specific epigenetic marks for MAR-containing genes vs. non-MAR genes, and comparison of gene expression levels. Gene expression is measured in RPKM, which probably needs to be clarified in the figure legend (it is defined in the text only). I'm not fully sure where these numbers are coming from. I assume from RNA-seq data? The methods do not reveal any details on this.

9) The AHL22-FRS7/12 complex is required for MARs attaching at the nuclear matrix – The authors present a comparison of MAR enrichment peaks over genes between the wildtype and triple mutant as evidence (figure 5). However, I think their conclusion may be a bit too strong. While the peaks are certainly lower in the mutant, they are not lost completely, suggesting that MARs are still enriched but to a lesser extent than in the wildtype. This suggests that there are also other mechanisms for MAR attachment besides AHL22-FRS7/12, so I would hesitate to use the word required. Also, FDR needs to be defined (False Discovery Rate?).

10) Small genes appear to have more MARs – This was delegated to the supplemental figures but is an interesting result that seemed unexpected at first. The authors get back to this later by pointing out that SAUR genes are short and thus subject to significant decrease of MAR enrichment in the mutant so it does fit into the overall picture.

11) Lower MAR attachment might cause transcriptional repression – The authors tested this by comparing expression levels of genes with decreased MAR enrichment vs. unaltered MAR enrichment in the triple mutant and wildtype. However, they found both up-regulated and down-regulated genes, suggesting that AHL22-FRS7/12-mediated MAR attachment is associated with both gene activation and repression.

12) SAUR genes are upregulated while having less MAR enrichment in the triple mutant compared to wildtype – This upregulation of auxin-response genes helps explain the long hypocotyl phenotype of the triple mutant. The authors validated the MAR enrichment over these genes using either RT-qPCR (text) or MAR-qPCR (figure legend). This needs clarification on the method used (I think it's MAR-qPCR). The overall conclusion then is that in the wildtype situation, the AHL22-FRS7/12 complex facilitates MAR attachment of these auxin-response genes, which leads to their suppression and thus inhibits hypocotyl elongation.

13) AHL22 interacts with histone deacetylase – The authors selected histone deacetylase HDA15, which has a gene silencing effect and has a similar mutant phenotype to the triple mutant, as a candidate interactor of AHL22 to test (figure 6). They were able to demonstrate interaction using BiFC and FRET-FLIM, suggesting that AHL22 is capable of recruiting HDA15 into the complex.

14) AHL22-FRS7/12 and HDA15 likely act together to co-repress auxin response genes – This was corroborated by comparing transcriptome profiles of the triple mutant with the *hda15* mutant and finding significant overlap. The authors used RT-qPCR to confirm the RNA-seq data for the SAUR gene family genes and found their expression was significantly higher in both the triple mutant and the *hda15* mutant compared to wildtype (figure 7). This supported the conclusion that in the wildtype situation, the interaction between AHL22-FRS7/12 and HDA15 conversely represses auxin signaling and thus hypocotyl elongation.

15) AHL22-FRS7/12 acts by recruiting HDA15 to remove H3 acetylation to repress auxin response genes – The authors first tested this with an immunoblot but did not see evidence of increased histone acetylation in the triple mutant. They then performed an H3ac enrichment using ChIP-qPCR and were able to show a significant increase of H3ac at the promoters of SAUR genes in the mutants compared to wildtype, along with a decrease in HDA15 recruitment in the triple mutant. This further supports their conclusion that AHL22-FRS7/12 represses hypocotyl elongation by

bringing together MARs and histone deacetylases to silence target genes. The authors present a working model in figure 7E. However, this part of the figure is never referred to anywhere in the text. It should probably be pointed out in the discussion.

Overall, I think this paper has a nice story and I particularly liked how the authors present their hypotheses and then lay out the experiments they used to test each one and what conclusion they were drawing from the results. I felt some of the points supported by supplementary data only may be better placed in the discussion (e.g. gene length). The entire paper could benefit from a spellcheck and proof-reading for consistency in wording. Some clarifications are probably needed to make things clearer for the audience, and a few things should be added such as more info about how the triple mutant was generated.

Response to review comments

Reviewer #1 (Remarks to the Author):

Comments: The story is very clear, but the key point still needs confirmation whether chromatin attachment to the nuclear matrix really happens first and subsequently it leads to repress hypocotyl elongation. The cause and effect relationship is not clear. There is still possibility that seedlings show hypocotyl elongation, then AHL22/FRS7/FRS12 would interact with each other and epigenetic changes could be observed as an effect. I would like to know how the authors could reply to the question.

Answer: Thank you very much for the comments. We appreciate that you consider our manuscript a clear story. In the manuscript, we provide several lines of evidence to support the proposed model:

(1) We found that AHL22 interacts with FRS7 and FRS12, and these proteins were identified in the results of proteomics of the nuclear matrix. This suggests that AHL22, FRS7, and FRS12 are nuclear matrix-associated proteins and interact with each other. This association provides a strong indication that these proteins play a role in nuclear matrix-related processes.

(2) The mutant that lacks AHL22, *FRS7*, and *FRS12* exhibited an elongated hypocotyl. This observation strongly suggests that these three proteins are required for the suppression of hypocotyl elongation. Hence, these proteins are involved in regulating the growth of the hypocotyl, rather than as a response to hypocotyl elongation.

(3) Our study also revealed a decrease in chromatin attachment at the nuclear matrix in the *ahl22 frs7 frs12* mutant, particularly at several auxin-related *SAUR* loci. Moreover, inhibition of auxin transportation could efficiently suppress the long hypocotyl in the mutant. This finding establishes a clear link between the attachment of chromatin to the nuclear matrix and the regulation of hypocotyl growth. The decrease in chromatin attachment in the mutant strongly supports the model that AHL22, FRS7, and FRS12 are essential for this process.

(4) In investigating the molecular mechanism, we discovered that AHL22 interacts with HDA15. The *hda15* mutant displayed a phenotype similar to the *ahl22frs7frs12* mutant. This genetic evidence further strengthens the connection between AHL22 and HDA15 in regulating hypocotyl growth. Additionally, the molecular analysis, including changes in H3 acetylation at the *SAUR* loci and HDA15 binding in the *ahl22frs7frs12* mutant, indicates that AHL22 recruits HDA15 to remove H3 acetylation, a key epigenetic modification associated with gene expression.

We believe that the above data altogether strongly supports the notion that these proteins are involved in the suppression of hypocotyl growth through their interactions, chromatin attachment at the nuclear matrix, and the recruitment of HDA15 for epigenetic modifications. Therefore, based on the genetics and the above other evidence, our current model seems the best possible model.

Comments: To substantiate the model more conclusively: Is it possible to make artificial/inducible tethering of AHL22/FRS7/FRS12 to matrix attachment regions to see subsequent seedling growth?

Answer: Thank you for your insightful comments. We appreciate your suggestion of artificially tethering AHL22, FRS7, and FRS12 to matrix attachment regions to investigate their impact on hypocotyl elongation. While we acknowledge the potential value of this approach, we would like to address that we currently don't have the system to implement the artificial tethering of AHL22, FRS7, and FRS12 to matrix attachment regions. Establishing such a system would indeed be a time-consuming endeavor. Moreover, according to our result, AHL22/FRS7/FRS12 inhibit hypocotyl elongation by regulating the expression of multiple SAURs. Hence, it potentially need tether AHL22/FRS7/FRS12 to multiple SAUR loci, if we want to test the effect of nuclear matrix attachment in hypocotyl elongation. In light of these considerations, we acknowledge your suggestion again and

concur that this is a worthwhile long-term research goal, and we have also included this suggestion in the discussion part (line 311-313).

Comments: Line 77: Why did the authors start AHL22 as the key factor in this study? Readers could not understand what is AHL22 and why AHL22 in the first place.

Answer: Thanks for the comment. When initiating this project, our primary objective was to investigate the role of AHL family proteins in regulating plant development through MAR attachment and epigenetic regulation. AHL22, specifically, emerged as a key focus due to its previously documented impact on hypocotyl growth and flowering time, potentially through histone modification¹. Additionally, the observed longer hypocotyl resulting from the mutation of AHL22 in conjunction with other AHLs further justified its selection as a target for our study². In response to your comments, we have enhanced the introduction section by incorporating this information at lines 70 and 76. We hope these additions provide better clarity regarding our rationale for choosing AHL22 as a central factor in our investigation.

Comments: Table 1: Out of these proteins, why did the authors start the study with AHL22, FRS7 and FRS12, but not with AHL27 or AHL29?

Answer: Thanks for the comments. We acknowledge the potential significance of exploring the functions of AHL27 and AHL29 in MAR attachment and epigenetic regulation. However, our study's narrative commenced with AHL22, and thus, it logically progressed to further investigate AHL22's role in these processes. The association between different AHL proteins has been previously reported^{3,4}, so we didn't continue in these associations.

FRS7 and FRS12, previously recognized chromatin-related proteins binding to AT-rich sequences⁵, were chosen as promising candidates. It's worth noting that while their chromatin interactions were known, their association with the nuclear matrix had not been examined until our study. Therefore, we chose AHL22, FRS7, and FRS12 as the initial proteins of interest in our investigation.

Comments: Fig.1C: Some differences were observed between AHL22-FRS7 and AHL22-FRS12. Does it not make biological significance? + (mean) marks are hard to see.

Answer: Thanks for the comments. We agree with the reviewer that there appears to be a difference in life time between AHL22-FRS7 and AHL22-FRS12 in figure 1C. The FLIM analysis was done to test the interaction between AHL22 and FRS7 or FRS12 and the obtained values clearly differ from the controls for both combinations, indicating that AHL22 interacts with both FRS isoforms. There is also a difference in the interaction strength according to the orientation of the tag on FRS, indicating a preferred orientation of the interaction. The experiments were performed separately in time and compared to their respective controls. It is theoretically possible that, at the structural level, there is a difference in the position of the fluorophores when associated to FRS7 or FRS12 and their distance to the donor, which can account for the minor difference in life time observed when comparing the AHL22-FRS7 and the AHL22-FRS12 pairs. However, as the experiments were not performed at the same time, the observed defects might also relate to minor differences in microscopy settings. At this point, and because the experiments were not performed at the same time, we do not find it appropriate to emphasize or speculate on the nature of these minor differences based on the current set of experiments.

We have changed the color of mean marks to red, which is easy to see now.

Comments: Abbreviations UG and DG firstly appeared in lines 127 and 129 without definition. The definition appeared later in line 212.

Answer: Thanks for the comments. We have moved the definition of the down-regulated genes (DGs) and up-regulated genes (UGs) to the first place of the manuscript, line 142.

Comments: Fig.4C: Epigenetic features should be respectively defined.

Answer: Thanks for the comments. We have included the following information in the legend: Distribution of average epigenetic features across MARs over genes. tri-methylation of histone H3 at lysine 4 (H3K4me3), tri-methylation of histone H3 at lysine 36 (H3K36me3), acetylation of histone H3 at lysine 9 (H3K9ac), acetylation of histone H3 at lysine 23 (H3K23ac), tri-methylation of histone H3 at lysine 27 (H3K27me3), mono-methylation of histone H3 at lysine 27 (H3K27me1), di-methylation of histone H3 at lysine 9 (H3K9me2), DNA methylation (DNAm), histone H3.1 (H3.1), histone H3.3 (H3.3), DNase-seq (DH), Pol2 ChIP-seq peaks (Pol2).

Reviewer #2 (Remarks to the Author):

In this study, Linhao and colleagues investigated the role of chromatin attachment to the nuclear matrix in regulating hypocotyl elongation in *Arabidopsis thaliana*. They identified a nuclear matrix binding protein, AHL22, that interacts with transcriptional repressors FRS7 and FRS12 to suppress the expression of a group of genes known as SAURs. The repression of SAURs depends on their attachment to the nuclear matrix. The AHL22 complex recruits the histone deacetylase HDA15 to the SAUR loci, leading to the removal of H3 acetylation and the suppression of hypocotyl elongation. This study reveals a novel mechanism by which nuclear matrix attachment to chromatin regulates gene expression and hypocotyl elongation. In principle, this work could be potentially very interesting as it provides (to my knowledge) the first characterization of the nuclear matrix and MARs in plants, while functionally linking its role to gene regulation. These results could have broad significance for research in plants however, I have a few questions would the work as it stands.

Comments: 1. My primary concern relates to the protocol used for MARs isolation, as this impacts most of the results in the study. The authors used a protocol from *Drosophila*, but how can we be certain that the composition is the same and that the protocol will work equally well in plants? In figure S1 the authors show images of nuclei at different steps of the protocol but I find them strange in appearance. For instance, no chromocenters are visible on the DAPI channel (instead, there are some rounded bodies that stain less with DAPI, which are also visible in brightfield images) and the nucleus size is too large (almost 50um in diameter). Also, after DNase I digestion no nuclei should be visible but in FigS1 the input image and after DNaseI digestion look similar. This leads me to question the accuracy of their protocol. Could the authors clarify this and maybe also try their protocol on AHL22-GFP lines, observing loss of DNA but presence of AHL22 after DNaseI digestion?

Answer: Thank you for your comments, and we appreciate your concerns regarding the MAR isolation protocol. We recognize the importance of ensuring the applicability of the protocol, especially when adapting it from *Drosophila* to plants. We followed a protocol that involved cutting DNA into fragments through enzymatic digestion, with subsequent removal of non-relevant proteins by high salt buffer washes. In the results of *Arabidopsis* MAR-seq, we could see that MARs are enriched surrounding transcription start sites, which pattern is very similar to the results in *Drosophila*⁶, suggesting the protocol works in *Arabidopsis* like in *Drosophila*. Moreover, a similar nuclear matrix isolation approach

containing high salt wash has been employed in the analysis of MAR attachment at the single locus in Arabidopsis ^{7,8}.

To address concerns related to the images of nuclei presented in Figure S1, we have obtained new photos illustrating three stages of the nuclear matrix isolation process. The nuclei's diameter measures approximately 5-10 μm , consistent with our expectations for Arabidopsis nuclei. As you correctly pointed out, we have identified chromosome centers within the isolated nuclei. Furthermore, as per your comments, we have observed a notable reduction in DAPI staining following DNase I digestion. Following the high salt buffer wash, the DAPI signal is not distinctly visible under the microscope. When searching for DNase I treated and high salt buffer washed nuclei, we couldn't find normal nuclei with normal chromosome centers. However, we could still identify the areas of nucleoli and some residual DAPI signal in DNase I treated nuclei, indicating their nature as nuclei. Similar effect was reported previously ⁹. All the aforementioned results support the successful isolation of Arabidopsis nuclear matrix with our protocol. In addition to the microscopic evidence, the results in Figure S1A also demonstrate a substantial decrease in histone proteins after the high salt buffer wash, further substantiating the notion that the high salt wash effectively removed non-relevant DNAs and proteins.

Regarding the experiment with AHL22-GFP lines, we appreciate your suggestion. While we attempted the suggested experiment, the weak expression of AHL22-GFP posed challenges in detecting the GFP signal even only after nuclei extraction. Consequently, we are failed to draw definitive conclusions about the effect of the nuclear matrix with the AHL22-GFP line after nuclear matrix isolation.

Comments: 2. The nuclear matrix was first reported in the 70s but since then numerous studies have failed to provide evidence for its existence and there's still some debate in the field. I think the authors should at least make a reference to that. I also felt that generally more introduction to nuclear matrix and MARs is necessary – for instance what is known in plants and why a nuclear matrix is necessary. In fact, this last point is not very well discussed. It is not clear to me what is the relevance of recruitment to nuclear matrix (especially since no correlation with gene expression or repression was observed)? For instance, could SAURs be recruited at the same nuclear location for co-regulation? I understand this may be beyond the scope of this paper but I felt like more discussion on this is needed.

Answer: Thanks a lot for the comments. We have included more background in the introduction (line 34-40, and line 48-53) and also discussion (line 314-319).

Comments: 3. The authors observed that AHL22-FRS7-FRS12 is required for MARs attaching to the nuclear matrix and that FRS proteins may function as MAR-binding proteins. To further support this claim I think it would be important to compare the genome-wide MAR attachment profiles in *ahl22* single mutant alone with the triple mutant and Col-0.

Answer: Thank you for your comments. Arabidopsis has 29 AHL proteins, and multiple AHL proteins in Arabidopsis have been reported as MAR-associated proteins ². The function of Arabidopsis AHL proteins in seedlings are highly redundant. The phenotype in hypocotyl growth only could be visible in high order mutants, dominant-negative line, or overexpression lines ². In line with previous publication, we did not see phenotype of hypocotyl growth in the *ahl22* mutant but only in *ahl22 frs7 frs12* mutant. Hence, we believe it is very possible that the single mutant has a similar MAR attachment pattern like in WT, and we decided to use WT and *ahl22 frs7 frs12* in the MAR-seq experiment.

FRS7 and FRS12 could interact with AHL22, and both of them are listed as MAR-associated proteins in the proteomic analysis of MAR-associated proteins. Hence, we believe FRS7 and FRS12 are also MAR-associated proteins. Accordingly, we could see that FRS7/12 also targets to AT-rich sequence in the

ChIP-seq analysis, which pattern is similar to other MAR-associated proteins, further supporting the idea that FRS7 and FRS12 are MAR-associated proteins. At last, in the regulation of hypocotyl growth, a well-known phenotype that is regulated by a group of AHL proteins by MAR-attachment, FRS7 and FRS12 act together with AHL22, again enhancing the idea that FRS7 and FRS12 are MAR associated proteins. At last, in the *ahl22 frs7 frs12* mutant that showed a long hypocotyl phenotype, we could see reduced MAR attachment associated with increased transcription of SAURs. Overall, we believe that our current evidence could support the idea that AHL22, FRS7, and FRS12 are MAR-associated proteins.

Comments: 4. Why HDA15 is not present on the nuclear matrix proteome or in the TAP-MS analysis of AHL22? It would be helpful if the authors could discuss this.

Answer: Thanks for the comments. Our proteomic analysis actually showed that multiple HDAs exist in the list of nuclear matrix-associated proteins in Arabidopsis, but we mistakenly not included in the manuscript. We have included the results in Figure S8 B now.

Comments:

Minor comments: • Citation missing: Igor V. Tetko et al., 2006 Plos Computational Biology. Showing evidence for nuclear matrix importance in plants.

- Line 98: salt instead of self.
- Line 464 and 467 Fig.S1 instead of Fig.S14.
- Figure 3: missing label on x-axis of graph (3B) – triple mutant.
- BiFC is also a method to test for protein interaction in planta. However, FRET/FLIM is presented as the only method doing so. Correct this miss reference.

Answer: Thank you very much for your comments. We have corrected and updated the above points in the revised manuscript.

Reviewer #3 (Remarks to the Author):

In this paper “Chromatin attachment to the nuclear matrix represses hypocotyl elongation in Arabidopsis thaliana” the authors present evidence that a nuclear matrix binding protein physically interacts with two transcriptional repressors at matrix attachment regions and regulates hypocotyl elongation by recruiting histone deacetylase to suppress the expression of auxin-response genes. I found this to be an intriguing story that ties gene regulation during plant development to nuclear matrix attachment and the concept of proteins exerting their function by acting as scaffolds for others to assemble into complexes.

As I was reading the paper, I noted a few comments/questions on each of the sections. 1) AHL22 interacts with FRS7 and FRS12 – This is demonstrated thoroughly by several methods including tandem affinity purification, followed by BiFC and FRET-FLIM, and co-immunoprecipitation (table 1, figure 1). The methods sections for these experiments in particular could benefit from a proof-reading and some clarification. E.g. the 35S promoter is from cauliflower (not tobacco) mosaic virus, OD=0.8 should probably specify the wavelength, and there was a reference missing listed simply by its PMID number. I noticed several typos, such as quantitative, combinations, and intensity that would all be easily caught by a spellcheck. The BiFC section mentions that H3.3-RFP was used as a marker protein but does not specify its source. It also mentions that percentage indicates strength of protein interactions, but this isn't included in figure 1. It is included later in figure 6D, which makes me wonder why it was omitted from figure 1.

Answer: Thank you for your valuable feedback. We appreciate your keen observations and have addressed the identified issues in the methods section. We have rectified the typos, clarified details

such as the source of the 35S promoter (cauliflower mosaic virus), specified the wavelength for OD=0.8, and included the missing reference.

Concerning the BiFC experiments, we acknowledge that quantitative analysis was not conducted for AHL22 and FRS7, as well as AHL22 and FRS12. Given that these interactions were robustly confirmed through multiple methods, we deemed quantitative analysis for BiFC unnecessary in these instances. However, in the case of AHL22 and HDA15, where only BiFC data was available, we opted for quantitative analysis to ensure the reliability of the BiFC results.

2) This interaction happens at the nuclear matrix – This is inferred from identifying all three proteins via LS-MS in isolated nuclear matrix fractions (figure 2). Nuclear matrix was isolated by high concentration salt solution (typo self in line 98 needs fixing) and resulted in several hundred proteins being identified including some conserved nuclear matrix proteins such as SUN1 and SUN2. I checked the data table for some other known nuclear matrix constituents and found them missing (e.g. CRWN1/LINC1). Some matrix proteins are notoriously difficult to solubilize and may not have been detectable by LS-MS. Thus, I think the sentence in the discussion claiming that this “allowed the identification of the composition of the Arabidopsis nuclear matrix” is probably overly optimistic and I would suggest changing that to say it allowed for identification of components of the nuclear matrix. In the proteome analysis section, the authors mention removing cytoplasmic proteins considered background, which makes me wonder how many of these were removed compared to the total. Importantly though all three proteins that were subject of the study cofractionated, corroborating the conclusion that FRS7 and FRS12 are likely nuclear matrix proteins that can interact with AHL22 at the nuclear matrix. In the table in figure 2 listing the AHL and FRS-related proteins identified, I thought the last column showing 3 replicates for everything was not needed and just adding visual clutter as it could be easily mentioned in the legend.

Answer: Thank you for your insightful comments. In the discussion section, we have moderated the language to accurately reflect the identification of components within the Arabidopsis nuclear matrix rather than claiming a comprehensive understanding of its entire composition.

In response to your query about the removal of cytoplasmic proteins, we have addressed this concern by specifying the number of removed cytoplasmic proteins in the manuscript (line 494). The occurrence of cytoplasmic proteins is very possible because the use of centrifugation to isolate nuclei during the nuclear matrix extraction process, which may also co-precipitate cytoplasmic organelles. Consequently, we identified 2872 cytoplasmic proteins associated with chloroplasts, mitochondria, or ribosomes, which were subsequently excluded from the list of nuclear matrix-associated proteins. At last, when checking the existence of CRWN1 in the nuclear matrix, we realized that we made a mistake when we filtered the dataset in Excel. The amount of identified nuclear matrix-associated proteins should be 1059. We have corrected this mistake in the manuscript and update the information in Table S2. And we also found that CRWN1 is in the current final list, which is what we could expect.

Additionally, we have revised the format of Figure 2 according to your suggestion, removing the unnecessary last column to enhance clarity and reduce visual clutter.

3) AHL22 cooperates with FRS7 and FRS12 to repress hypocotyl elongation – This is demonstrated by the phenotypes of mutant lines with a triple mutant showing significantly longer hypocotyls compared to single or double mutants, suggesting genetic interaction (figure 3). Conversely, overexpressing lines showed shorter hypocotyls. The methods section lists the sources for the single and double mutants, as well as the overexpressing lines, but makes no mention of how the triple mutant was generated, nor is there any experimental evidence provided confirming its triple mutant status. I’m assuming the

triple mutant line was generated by the authors using genetic crosses and PCR for genotyping, but that info is missing and should be added.

Answer: Thank you for bringing this to our attention. To generate the triple mutant line (*ahl22 frs7 frs12*), we performed a cross between the *ahl22* mutant and the *frs7 frs12* double mutant. The confirmation of the triple mutant status was conducted through genotyping, utilizing primers in Table S6. We have now incorporated these details into the "Plant materials and growth conditions" section of the Materials and Methods.

4) The mode of repression is via regulating gene expression – Evidence for this is provided by comparing transcriptome profiles, presumably generated by RNA-seq listed in the methods (this might need clarification), and finding substantial overlap between differentially expressed genes in the mutants compared to wildtype. Further things I noted as needing clarification are defining UGs and DGs in line 127 (it's done in line 212 but should be done here when those abbreviations are first used) and the mention of a volcano plot in line 505. I did not see any volcano plots being used in either the paper or the supplemental figures.

Answer: Thanks for the comments. We have defined UGs and DGs at the first place, line 142 in the text. And, we have removed "volcano plot" from the manuscript.

5) This gene repression specifically targets the auxin pathway – This conclusion is backed up by finding higher expression levels of genes responsive to auxin in the mutant lines, and conversely an inhibitor of auxin transport suppressing the long hypocotyl phenotype in the mutants. That is nice data but I found the choice of graph in figure 3F a little puzzling. The line graph is suggestive of a sequence such as a timeline. I suspect it is meant to connect increasing levels of NPA but I think a bar graph would work better here. I also think the figure legend should define NPA (it's spelled out in the text but not in the figure), and I assume the DMSO is the solvent for the NPA and included as a control.

Answer: Thanks for the comments. We have updated the figure with a suggested bar graph here. And we have also included the description of NPA and DMSO in the legend.

6) AHLs and FRS7/12 possibly interact directly with the promoters of the target genes – The authors infer this from published datasets for related proteins that co-precipitated with AHL22 because they were not able to get data from the ChIP-seq experiment with AHL22. This is an obvious gap in the data but an argument can be made that these proteins interact as a complex and thus likely target the same genes.

Answer: Thank you for your understanding and consideration. We acknowledge the gap in the data regarding the ChIP-seq experiment with AHL22. To address this limitation, we utilized the ChIP-seq results of AHL29, and we appreciate your recognition of this approach.

7) MARs are enriched at TSS and TES of protein-coding genes – This is demonstrated by MAR-seq, which is sequencing DNA extracted from isolated nuclear matrix (figure 4). This appears to be a novel method developed in animals applied to plants. TSS should be defined where it is first used as an abbreviation. Another abbreviation that needs clarification is TEs vs. TES. It's not explained in either the text or the figure legend, but from the context it is clear TE is meant to contrast with protein-coding genes. Transposable elements? Also, the figure legend mentions aligning TTSSs, but there is nothing labeled TTS in the figure. It is called a metagene plot in the figure legend but metaplot in the bioinformatics section of the methods.

Answer: Thanks for the comments. We have defined TE in both text (line 182) and the legend, and we also replaced “TTS” with “TES” in the legend and “metaplot” with “metagene plot” in the section of the methods.

8) MAR enrichment correlates with active epigenetic marks of highly expressed genes – This is demonstrated with more metagene plots comparing the presence of specific epigenetic marks for MAR-containing genes vs. non-MAR genes, and comparison of gene expression levels. Gene expression is measured in RPKM, which probably needs to be clarified in the figure legend (it is defined in the text only). I’m not fully sure where these numbers are coming from. I assume from RNA-seq data? The methods do not reveal any details on this.

Answer: In the figure legend, we have expanded RPKM to its full name, "reads per kilobase of exon per million reads mapped," for better clarity. Furthermore, we have explicitly stated in the legend and the text (line 194) that the gene expression levels were obtained from RNA sequencing data. To provide additional information on RPKM calculation, we have mentioned that we used calculated RPKM with R in the "Bioinformatic analysis" section of the Methods.

9) The AHL22-FRS7/12 complex is required for MARs attaching at the nuclear matrix – The authors present a comparison of MAR enrichment peaks over genes between the wildtype and triple mutant as evidence (figure 5). However, I think their conclusion may be a bit too strong. While the peaks are certainly lower in the mutant, they are not lost completely, suggesting that MARs are still enriched but to a lesser extent than in the wildtype. This suggests that there are also other mechanisms for MAR attachment besides AHL22-FRS7/12, so I would hesitate to use the word required. Also, FDR needs to be defined (False Discovery Rate?).

Answer: Thank you for your thoughtful comments. We have adjusted our conclusion to more accurately reflect the data. The revised statement now indicates that the AHL22-FRS7/12 complex "is involved in MAR attachment regulation". Additionally, we have defined FDR (False Discovery Rate) at line 146 to provide clarity to readers.

10) Small genes appear to have more MARs – This was delegated to the supplemental figures but is an interesting result that seemed unexpected at first. The authors get back to this later by pointing out that SAUR genes are short and thus subject to significant decrease of MAR enrichment in the mutant so it does fit into the overall picture.

Answer: According to your last suggestion, we have moved this part to the discussion part (line 319-324).

11) Lower MAR attachment might cause transcriptional repression – The authors tested this by comparing expression levels of genes with decreased MAR enrichment vs. unaltered MAR enrichment in the triple mutant and wildtype. However, they found both up-regulated and down-regulated genes, suggesting that AHL22-FRS7/12-mediated MAR attachment is associated with both gene activation and repression.

Answer: We appreciate your feedback.

12) SAUR genes are upregulated while having less MAR enrichment in the triple mutant compared to wildtype – This upregulation of auxin-response genes helps explain the long hypocotyl phenotype of the triple mutant. The authors validated the MAR enrichment over these genes using either RT-qPCR

(text) or MAR-qPq2CR (figure legend). This needs clarification on the method used (I think it's MAR-qPCR). The overall conclusion then is that in the wildtype situation, the AHL22-FRS7/12 complex facilitates MAR attachment of these auxin-response genes, which leads to their suppression and thus inhibits hypocotyl elongation.

Answer: Thanks for the comments. We have corrected to MAR-qPCR in the text.

13) AHL22 interacts with histone deacetylase – The authors selected histone deacetylase HDA15, which has a gene silencing effect and has a similar mutant phenotype to the triple mutant, as a candidate interactor of AHL22 to test (figure 6). They were able to demonstrate interaction using BiFC and FRET-FLIM, suggesting that AHL22 is capable of recruiting HDA15 into the complex.

Answer: We appreciate your feedback.

14) AHL22-FRS7/12 and HDA15 likely act together to co-repress auxin response genes – This was corroborated by comparing transcriptome profiles of the triple mutant with the *hda15* mutant and finding significant overlap. The authors used RT-qPCR to confirm the RNA-seq data for the SAUR gene family genes and found their expression was significantly higher in both the triple mutant and the *hda15* mutant compared to wildtype (figure 7). This supported the conclusion that in the wildtype situation, the interaction between AHL22-FRS7/12 and HDA15 conversely represses auxin signaling and thus hypocotyl elongation.

Answer: We appreciate your feedback.

15) AHL22-FRS7/12 acts by recruiting HDA15 to remove H3 acetylation to repress auxin response genes – The authors first tested this with an immunoblot but did not see evidence of increased histone acetylation in the triple mutant. They then performed an H3ac enrichment using ChIP-qPCR and were able to show a significant increase of H3ac at the promoters of SAUR genes in the mutants compared to wildtype, along with a decrease in HDA15 recruitment in the triple mutant. This further supports their conclusion that AHL22-FRS7/12 represses hypocotyl elongation by bringing together MARs and histone deacetylases to silence target genes. The authors present a working model in figure 7E. However, this part of the figure is never referred to anywhere in the text. It should probably be pointed out in the discussion.

Answer: Thank you for your feedback. We have incorporated a reference to Figure 7E in both the results and discussion sections to ensure clarity and consistency in the presentation of our working model.

Overall, I think this paper has a nice story and I particularly liked how the authors present their hypotheses and then lay out the experiments they used to test each one and what conclusion they were drawing from the results. I felt some of the points supported by supplementary data only may be better placed in the discussion (e.g. gene length). The entire paper could benefit from a spellcheck and proof-reading for consistency in wording. Some clarifications are probably needed to make things clearer for the audience, and a few things should be added such as more info about how the triple mutant was generated.

Answer: Thank you for your thorough and constructive feedback. We appreciate your positive comments on the overall structure of the paper and the clarity in presenting hypotheses and experimental results. We'll certainly take your suggestion into consideration and consider moving

some supplementary data points to the discussion for better flow. Additionally, we acknowledge the importance of a thorough proofread and will ensure consistency in wording throughout the paper. We'll work on providing more details, particularly on the generation of the triple mutant, to enhance the transparency of our methodology.

Reference

1. Yun J, Kim YS, Jung JH, Seo PJ, Park CM. The AT-hook motif-containing protein AHL22 regulates flowering initiation by modifying FLOWERING LOCUS T chromatin in Arabidopsis. *J Biol Chem* **287**, 15307-15316 (2012).
2. Zhao J, Favero DS, Peng H, Neff MM. Arabidopsis thaliana AHL family modulates hypocotyl growth redundantly by interacting with each other via the PPC/DUF296 domain. *Proc Natl Acad Sci U S A* **110**, E4688-4697 (2013).
3. Favero DS, *et al.* SUPPRESSOR OF PHYTOCHROME B4-#3 Represses Genes Associated with Auxin Signaling to Modulate Hypocotyl Growth. *Plant Physiol* **171**, 2701-2716 (2016).
4. Jiang H, *et al.* Ectopic application of the repressive histone modification H3K9me2 establishes post-zygotic reproductive isolation in Arabidopsis thaliana. *Genes Dev*, (2017).
5. Ritter A, *et al.* The transcriptional repressor complex FRS7-FRS12 regulates flowering time and growth in Arabidopsis. *Nat Commun* **8**, 15235 (2017).
6. Rudd S, Frisch M, Grote K, Meyers BC, Mayer K, Werner T. Genome-wide in silico mapping of scaffold/matrix attachment regions in Arabidopsis suggests correlation of intragenic scaffold/matrix attachment regions with gene expression. *Plant Physiol* **135**, 715-722 (2004).
7. Ng KH, Yu H, Ito T. AGAMOUS controls GIANT KILLER, a multifunctional chromatin modifier in reproductive organ patterning and differentiation. *PLoS Biol* **7**, e1000251 (2009).
8. Xu YF, *et al.* A Matrix Protein Silences Transposons and Repeats through Interaction with Retinoblastoma-Associated Proteins. *Curr Biol* **23**, 345-350 (2013).
9. Dobson JR, *et al.* Identifying Nuclear Matrix-Attached DNA Across the Genome. *J Cell Physiol* **232**, 1295-1305 (2017).

Reviewer #1 (Remarks to the Author):

After being revised by the authors, I am satisfied with the revised draft and authors' responses to all the three reviewers. In the beginning the authors put some historical background before start the following core part. The reviewers all evaluated its significance, making a kind of pioneering studies in the field.

The authors also modified/added important method procedures following the reviewers' comments

Reviewer #2 (Remarks to the Author):

The authors have addressed all my comments and I am satisfied with most of their responses. I suggest that this manuscript is ready for publication.

Reviewer #3 (Remarks to the Author):

The authors addressed all major points in my previous review in the revised version of the paper "Chromatin attachment to the nuclear matrix represses hypocotyl elongation in *Arabidopsis thaliana*." I appreciate the update to the filtering of the Excel dataset to eliminate errors, additions to the methods section, and clarification of wording. The only minor detail mentioned that is still missing is the source for the H3.3-RFP used as a marker for nuclei (BiFC section in the methods, figure 6C).

Response to reviewer's comments

Reviewer #1:

After being revised by the authors, I am satisfied with the revised draft and authors' responses to all the three reviewers. In the beginning the authors put some historical background before start the following core part. The reviewers all evaluated its significance, making a kind of pioneering studies in the field.

The authors also modified/added important method procedures following the reviewers' comments

Reviewer #2:

The authors have addressed all my comments and I am satisfied with most of their responses. I suggest that this manuscript is ready for publication.

Reviewer #3:

The authors addressed all major points in my previous review in the revised version of the paper "Chromatin attachment to the nuclear matrix represses hypocotyl elongation in *Arabidopsis thaliana*." I appreciate the update to the filtering of the Excel dataset to eliminate errors, additions to the methods section, and clarification of wording. The only minor detail mentioned that is still missing is the source for the H3.3-RFP used as a marker for nuclei (BiFC section in the methods, figure 6C).

Answer: We apologize for missing the information of the H3.3-RFP plasmid, and we have included this information in the updated manuscript (line 363 and 364).